# Experimental realization of an extended Fermi-Hubbard model using a 2D lattice of dopant-based quantum dots

Xiqiao Wang [1,2,5], Ehsan Khatami [3], Fan Fei[1,4], Jonathan Wyrick[1], Pradeep Namboodiri[1], Ranjit Kashid[1,6], Albert F. Rigosi [1], Garnett Bryant [1,2] & Richard Silver [1] ✉

The Hubbard model is an essential tool for understanding many-body physics in condensed matter systems. Artificial lattices of dopants in silicon are a promising method for the analog quantum simulation of extended Fermi-Hubbard Hamiltonians in the strong interaction regime. However, complex atom-based device fabrication requirements have meant emulating a tunable two-dimensional Fermi-Hubbard Hamiltonian in silicon has not been achieved. Here, we fabricate 3 × 3 arrays of single/few-dopant quantum dots with finite disorder and demonstrate tuning of the electron ensemble using gates and probe the many-body states using quantum transport measurements. By controlling the lattice constants, we tune the hopping amplitude and long-range interactions and observe the finite-size analogue of a transition from metallic to Mott insulating behavior. We simulate thermally activated hopping and Hubbard band formation using increased temperatures. As atomically precise fabrication continues to improve, these results enable a new class of engineered artificial lattices to simulate interactive fermionic models.

Analog quantum simulators are designed quantum systems with a tunable Hamiltonian to emulate complex quantum systems intractable using classical computers due to the exponential growth of the Hilbert space with the system size[1]. Simulating strongly interacting fermions on a lattice lies at the heart of understanding quantum many-body phenomena, such as high-Tc superconductivity[2] and spin liquidity[3] that emerge in solid-state systems at low temperatures and are not describable through mean-field or density functional theory.

Various experimental platforms that form artificial lattices have been explored for realizing Fermi-Hubbard analog quantum simulators, including optical lattices[4,5], moiré superlattices[6], and semiconductor quantum dot systems[7,8]. Quantum dots, often referred to as artificial atoms, can be arranged into artificial molecules and lattices with tunable hopping amplitude, interaction strength, and custom-

designed point symmetry. For probing Fermionic many-body physics, the unique advantages of quantum-dot systems relative to other platforms, such as cold atoms in optical lattices, include a readily achievable low-temperature-limit with respect to the hopping amplitude, easy access to transport measurements, and dynamic control of the chemical potential landscape and filling factors using gates[9]. Amongst the various semiconductor quantum dot systems, lattices of dopant-based quantum dots have unique advantages in simulating strongly correlated Fermionic systems of real atomic lattice sites because the atomic nature of the quantum dots means they have naturally occurring ion-cores, nuclear spins, hyperfine interactions, and inherently strong long-range interactions. Additionally, patterning the device geometry using the scanning tunneling microscope (STM)-

[1]Atom Based Device Group, National Institute of Standards and Technology, Gaithersburg, MD 20899, USA. [2]Joint Quantum Institute, University of Maryland, College Park, MD 20740, USA. [3]Department of Physics and Astronomy, San José State University, San José, CA 95192, USA. [4]Department of Physics, University of Maryland, College Park, MD 20740, USA. [5]Present address: Rigetti Computing, Fremont, CA 94538, USA. [6]Present address: Center for Materials for Electronics Technology, Pune 411008, India. ✉e-mail: Richard.silver@nist.gov

based hydrogen lithography technique[10] adds the versatility of tailoring complex gate designs.

Effective control of tunable Hamiltonian parameters and precision-engineering of electron and spin correlations in a dopant-based artificial lattice relies on the controlled placement of dopant atoms in the host lattice with near-atomic precision. Although the Anderson-Mott transition has been previously demonstrated in few-atom systems using ion-implanted single dopant impurity atoms in silicon[11], this technique is limited by implantation aperture size and ion straggle. To our knowledge the best reported implant positioning accuracy is ~5–10 nm[11], which is incompatible with multiple site atomic-precision dopant arrays (In this study, we define "atomic-scale precision" as achieving sub-nm positioning accuracy of the artificial lattice sites using single/few-dopant quantum dots). The STM-based hydrogen lithography technique, initially pioneered by Joseph Lyding's group[12], was further developed and expanded by Michelle Simmons' group at UNSW[10,13] into a method that allows controlled placement of

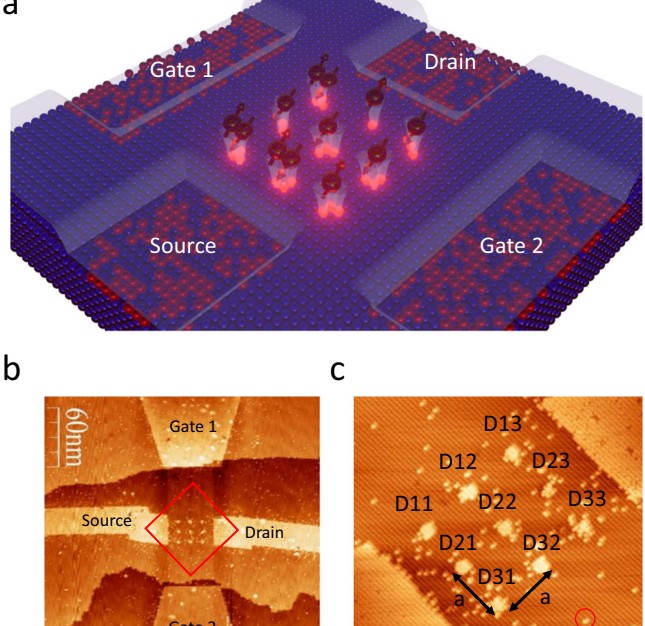

**Fig. 1 | 2D dopant-based quantum dot arrays as a platform for simulating the extended Fermi-Hubbard model. a** Schematic of the experimental Fermi-Hubbard system composed of a 3 × 3 array of single/few-dopant quantum dots coupled to in-plane gates and source-drain leads, allowing transport measurements through the array. The number of electrons (shown as arrows) and dopant atoms at each array site (pink dots) in this schematic have been arbitrarily assigned for illustration purposes. **b** STM image of the central device region of the 3 × 3 array acquired immediately following hydrogen lithography. The lithography patterns appear as bright regions on the hydrogen-terminated Si(100) surface due to hydrogen-depassivation and the exposure of chemically reactive Si dangling bonds. In this image, the central array and the source/drain leads reside on the same Si(100) surface terrace; Gate 1 and Gate 2 reside on surface terraces one monoatomic layer (~0.14 nm) above and below the middle terrace. **c** Atomic resolution STM image of the 3 × 3 array pattern (zoom of marked square region in **b**). Each dot is numbered to facilitate the discussion in the main text. In this array, $a = 10.7 \pm 0.3 nm$ is the square lattice constant that equals the average center-to-center distance between nearest neighboring sites. 2 × 1 surface reconstruction dimer rows on the Si(100) surface run from the upper left to lower right direction in the image. We define the 2D square lattice constant, a, as the averaged center-to-center distance between nearest neighboring dots within the array. The red circle marks an example of an isolated single dangling bond that does not incorporate dopant atoms. The STM image is taken at −2 V sample bias and 0.1 nA setpoint current.

dopant atoms buried in an epitaxial Si environment, enabling a new suite of atomic-scale precision devices. Several groups have further developed this atomic-scale device fabrication technique introducing new applications, e.g., NIST (dopant-based analog quantum simulation)[14,15], Sandia (CMOS integration)[16,17], and UCL,[18,19]. These advances have demonstrated success in atom-by-atom construction of single and few-dopant quantum dot devices in silicon and atomic-scale control of tunneling in dopant-based devices, enabling high-fidelity dopant-based multi-qubits[20] and simulation of the topological phases of the 1-dimensional many-body Su−Schrieffer−Heeger (SSH) model[21]. Here, we develop a path towards precise fabrication of dopant-based Fermi-Hubbard simulators where the on-site electron-electron interactions can be controlled by engineering the number and configuration of dopant atoms at each site, while the site-to-site hopping amplitude and strength of long-range interactions can be controlled by altering the spatial separations between sites. Limited theoretical studies have been carried out to predict the many-body properties in dopant-based quantum dot arrays in silicon[22–24], such as strong correlation, excitation spectrum, and their robustness against disorder for analog quantum simulation. Transport through many-body states in dopant-based arrays has also been proposed as a sensitive probe to topological phase transitions[24] and coherent manipulation of electronics states[25]. Until now, however, atomic-scale fabrication and quantum transport characterization of 2D artificial lattices of single/few-dopants have not been realized in the laboratory.

Here, we simulate an extended 2D Fermi-Hubbard Hamiltonian using atomic-scale fabricated 3 × 3 lattices of single/few-dopant quantum dot, although disorder present in the arrays leads to uncertainty in the underlying Hamiltonian. We define the electron ensemble in the array by tuning the chemical potential landscape using in-plane gates and measure the low-temperature quantum transport through the array to probe the charge addition spectrum, resonant tunneling, and the impact of inhomogeneity within the many-body system of the arrays. By increasing the average lattice constant from ~4.1 nm to ~10.7 nm, we tune the site-to-site hopping amplitude and long-range interactions within the simulated Hamiltonian and observe a transition from metallic behavior to the formation of Coulomb blockade. Finally, at elevated temperatures, we observe the formation of Hubbard bands in transport spectroscopy that can be attributed to additional hopping introduced by thermally activated occupation of many-body states. To augment the interpretation of our simulated extended Hubbard Hamiltonian, we numerically solve the Hubbard Hamiltonian using exact diagonalization and parameters estimated based on the experimental characterization of the array. Our results establish a new solid-state platform for the exploration of extended Fermi-Hubbard models of strongly correlated 2D systems.

## Results

### 3 × 3 arrays of few-dopant quantum dots and the extended Fermi-Hubbard Hamiltonian

We fabricated[14] a series of 3 × 3 square lattice arrays of few-dopant quantum dots that are weakly tunnel coupled to a source and drain and capacitively coupled to two in-plane gates. Figure 1 shows STM images of the hydrogen lithography patterns from one of the arrays (average lattice constant $a = 10.7 \pm 0.3 nm$) on a hydrogen-terminated Si(100) 2 × 1 reconstruction surface, where the locations of the artificial lattice sites and the lattice constants can be determined by using the surface reconstruction dimer unit cells as an atomically precise ruler and counting the number of dimer rows (dimer-row pitch = 0.77 nm) between neighboring sites. Each lattice site is defined by using an STM tip to remove a small patch of (~10 to ~20) adjacent hydrogen atoms, allowing individual phosphorus atoms to incorporate only into the exposed surface Si lattice sites in a subsequent phosphorus dosing and incorporation process. We have recently demonstrated that a dangling bond patch of similar size typically forms a few-dopant cluster

quantum dot incorporating 1 to 3 phosphorus atoms[14]. At the same time, the STM-patterned in-plane source/drain leads and two symmetric in-plane gates are saturation-doped to a dopant density of ~2×10$^{14}$/$cm^2$ that corresponds to a bulk doping density of 2×10$^{21}$/$cm^3$, approximately three orders of magnitude above the bulk metal-insulator transition, allowing quasi-metallic conduction in all electrodes. We use a room-temperature grown locking layer technique to suppress atomic-scale movement of the precisely defined dopant atom positions[26] before a subsequent low-temperature (~250 °C) epitaxial Si overgrowth, that embeds the dopant atoms in a 3-dimensional crystalline Si environment. Finally, ohmic contacts to the buried electrodes are formed using a low thermal budget silicide contacting technique[27].

The Hubbard model has long provided a theoretical playground for understanding different phases of matter, especially in the presence of strong electronic correlations. The 3 × 3 array's physical attributes map to an extended Fermi-Hubbard Hamiltonian, which, in its simplest form, includes one spinful orbital at each site. Experimentally, the absolute number of excess electrons at each single/few-dopant quantum dot may vary according to the inhomogeneity in the number of dopant atoms at each lattice site. For quantum transport through dopant-based quantum dots within the small bias ranges used in this study, charge fluctuations at quantum dots occur via quantum dot energy levels that are near or in between the Fermi levels in the source and drain leads. Therefore, we limit our analysis to on-site binding energy levels that are nearest to the Fermi level, i.e., charge number fluctuations of up to two electrons at each site which corresponds to the three charge states of a few-dopant quantum dot: the ionized state, the charge-neutral state, and the negatively charged state. The absolute number of excess electrons on a few-dopant quantum dot does not affect the underlying physical phenomena in this work.

$$H = H_\mu + H_t + H_U$$
$$H_\mu = \sum_{i\sigma} \mu_i n_{i\sigma} = \sum_{i\sigma}(p_i + E_{bi} + \sum_{j, i\neq j} V_{i,j})n_{i\sigma}$$
$$H_t = \sum_{\langle i,j\rangle,\sigma}(-tc_{i\sigma}^\dagger c_{j\sigma} + H.c.) \tag{1}$$
$$H_U = \sum_i U_i n_{i\uparrow} n_{i\downarrow} + \sum_{i,j, i\neq j} U_{ij} n_i n_j$$

The total Hamiltonian consists of the onsite energy terms in $H_\mu$, the kinetic energy (hopping) terms in $H_t$, and the electron-electron interaction energy terms in $H_U$. Here, $n_{i\sigma} = c_{i\sigma}^\dagger c_{i\sigma}$ is the number operator where $c_{i\sigma}^\dagger$ ($c_{i\sigma}$) is the creation (annihilation) operator of a fermion with spin $\sigma$ at lattice site $i$. $\mu_i$ is the chemical potential at site $i$ comprised of fixed contributions from local and long-range electron-ion core Coulomb interactions, i.e., $E_{bi}$ (electron binding energy at site $i$) and $V_{i,j}$ (Coulomb attraction between an electron at site $i$ and an ion-core at site $j$); $E_{bi}$ is determined by the number of dopants and detailed dopant-cluster configurations at each site. $V_{i,j}$ is determined by the separation between two lattice sites and can be approximated using a point-charge approximation $V_{i,j} \approx -\frac{V_0}{|R_i - R_j|}$ where $V_0 = \frac{e^2}{4\pi\varepsilon_r\varepsilon_0} \approx 123meV \cdot nm$ in silicon (see Supplementary Note 1). $\mu_i$ also includes a tunable contribution $p_i$ that is determined by the classical capacitance couplings of the device and the applied voltages on the gates and source/drain leads. $t$ represents the hopping amplitude between nearest-neighbor sites ($H.c.$ indicates Hermitian conjugate). $U_i$ is the local electron-electron Coulomb repulsion at site $i$, and $U_{ij}$ is the long-range Coulomb repulsion between electrons at sites $i$ and $j$. The filling factor (electron number) in the array is determined by the position of the array's chemical potential with respect to the Fermi levels in the source/drain leads, which we set as $E_F = 80meV$ below the Si conduction band edge[10]. We ignore Coulomb exchange and higher-order hopping terms in our numerical model[28].

The hopping $t$, interactions $U_i$, $U_{i,j}$, chemical potential terms $\mu_i$, which determines the electron distribution and doping level of the array, and the temperature $T$ constitute the Fermi-Hubbard model's parameter space, covering a large variety of correlated electron phenomena, some not yet fully understood, and others that continue to be discovered. Physical control of the Hubbard model parameters in our 2D arrays is achieved by varying device fabrication and measurement conditions. The number of electrons in the array ($\sum n_i$), can be altered by applying a common voltage on both in-plane gates to shift the array's chemical potential landscape (albeit non-uniformly due to screening from source and drain leads) with respect to the Fermi level. A voltage difference between the two gates will introduce a potential gradient within the 2D array along the gate-gate direction. In contrast to gate-defined quantum dot systems, in-situ gate tuning of tunnel coupling within a single device is less efficient in donor-defined quantum dot systems. Effects of tuning the hopping amplitude $t$ and long-range interaction terms ($U_{i,j}$) can be achieved, however, by fabricating a series of dopant-based lattices with different lattice constants that are determined at the fabrication stage. The local interaction term $U_i$ reflects the physical size of each quantum dot. For the arrays in this study, we design the in-plane gates and average lithographic dot size to be the same for all arrays and only alter the lattice constant from $a_1 = 4.1 \pm 0.3nm$ in the first array to $a_2 = 6.6 \pm 0.3nm$ in the second array and to $a_3 = 10.7 \pm 0.3nm$ in the third array. Based on previous theoretical studies[28,29], these lattice constants correspond to hopping amplitudes in the ~8 $meV$ to the hundreds of $\mu eV$ range and long-range interactions in the range of ~20 meV to few-$meV$ (See Supplementary Note 1). We estimate the number of dopant atoms in the quantum dots to be within the range of $2 \pm 1$ dopants by characterizing the binding energies of few-dopant quantum dots with a similar lithographic patch (See Supplementary Notes 2 and 3). A quantum dot size of $2 \pm 1$ dopants corresponds to a local electron-electron interaction energy $U_i$ of ~45 meV (for the relevant charge-neutral to negatively charged state transitions). These energy scales position these Hubbard arrays in the strongly interacting regime with non-negligible long-range interactions, even beyond nearest-neighbor sites[30]. Our best estimates point to an average ratio $U_i/t$ that varies roughly from 6 to 90 from the first to the third array. The ratio of the nearest-neighbor Coulomb repulsion and the hopping amplitude also varies by an order of magnitude roughly from 2 to 20. So, while we expect Coulomb interactions to dominate in the third array, their strengths are comparable to, or less than, the non-interacting bandwidth (see SM, Fig. S4) in the metallic array, and therefore, we expect the significant tunneling/delocalization of electrons to change the character of the latter system. Disorder also plays a significant role in all three arrays. Apart from introducing site-to-site variations in interactions and tunneling amplitudes, the ratio of its strength (defined as the difference in binding energies for 1 and 3 dopants per site) and the hopping amplitudes varies roughly from 4 to 70 from the first to the third array.

Atomic-scale defects have been a critical challenge for solid-state implementations of quantum devices that rely on atomic precision fabrication processes such as those used here. The primary source of defect/disorder is the site-by-site variation in on-site energies $E_{bi}$ and $U_i$, which effectively introduces inhomogeneity in the chemical potential landscape, interactions, and hopping amplitudes. Due to the stochastic nature of the phosphorus dosing and incorporation process[31], deterministic control of the exact number of dopant atoms and their specific cluster configuration within a lithographic patch remains an unsolved challenge in the community[16]. While the precise atomic configurations in an array may, in principle, be obtained by parametrically fitting the measurement results with numerical simulations, a detailed disorder configuration investigation is extremely computationally expensive and beyond the scope of this study. Instead, we account for disorder by estimating the number of dopant atoms-per-site, based on STM-lithography patterns

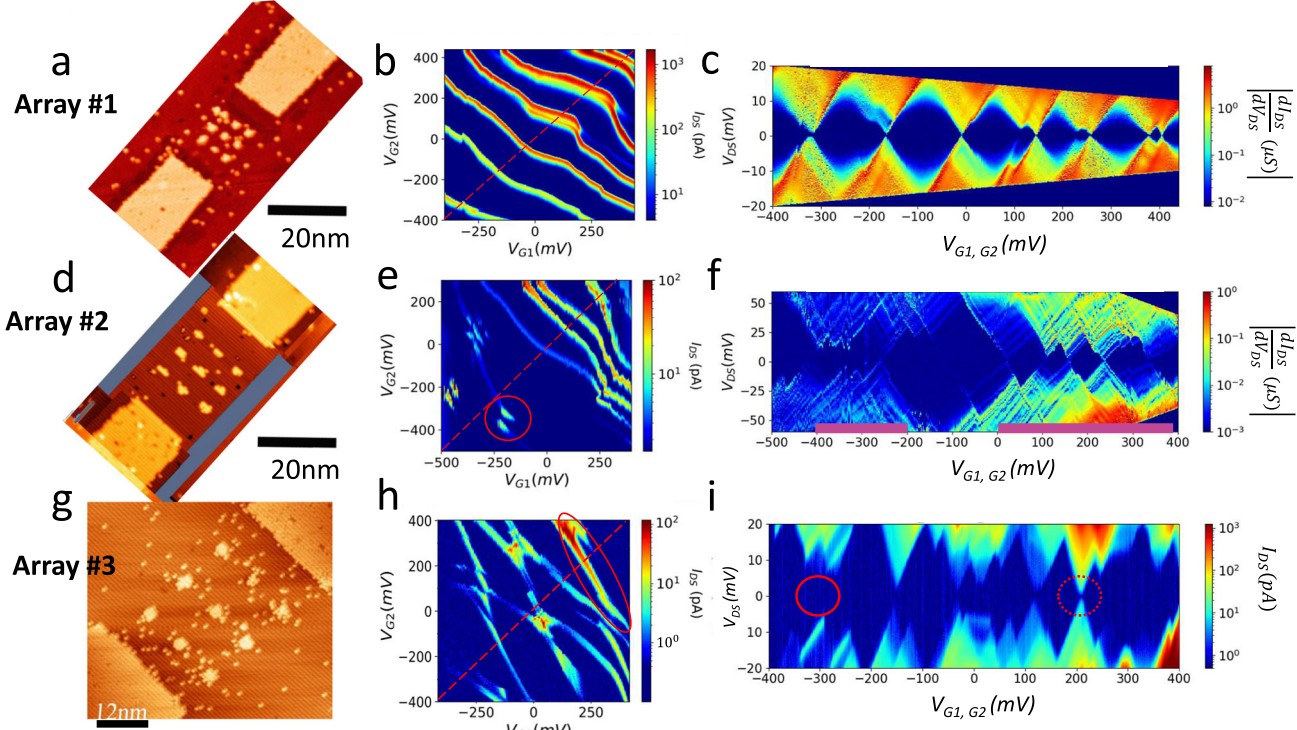

**Fig. 2 | Transition from metallic behavior at reduced lattice constant to a weakly tunnel coupled array in the Coulomb blockade regime. a, d, g** are STM images of the three arrays, where the lattice constants vary from $a_1 = 4.1 \pm 0.3 nm$, $a_2 = 6.6 \pm 0.3 nm$ to $a_3 = 10.7 \pm 0.3 nm$, respectively. **b, e, h** Experimentally measured DC conductance charge stability diagrams of the three arrays with $V_{bias} = 2 mV$ for array one and at $V_{bias} = 4 mV$ for arrays two and three at the base temperature of the

dilution refrigerator ($T = 10 mK$). **c, f, i** Differential conductance Coulomb blockade diagrams for arrays one and two and DC conductance Coulomb blockade diagram for array three measured at base temperature along the dashed lines in **b, e, h**, respectively. Solid and dashed circles in (**i**) mark examples of open and closed Coulomb diamonds that are described in the main text.

and dopant incorporation conditions[32], for use as input to the numerical simulations. We do not distinguish the effects from different sources of disorder because variation in dopant number per site and their clustering configuration have similar effects on site-by-site energy variations. While we have not pursued the exact match between the numerically simulated and experimentally simulated results in this study, the detailed atomic configuration at each array site does not alter the qualitative understanding of the array system (see Supplementary Note 6), and the quantitative differences between theory and experiment are a measure of the accuracy of the disorder estimates. Supplementary Table 3 in the Supplementary Notes lists the range of binding energy $E_b$ and the on-site addition energy $U_i$ for 1 P, 2 P, and 3 P dopant clusters[29]. Variations in the number of dopant atoms and detailed dopant cluster configurations at each lattice site introduce site disorder, which in larger arrays lends itself to the study of Anderson/Mott localization, and especially its existence in the presence of strong interactions[27].

In the following sections, we demonstrate the tunability of dopant-based 2D arrays by first showing tuning of the hopping amplitudes and long-range interactions through transport measurements of arrays fabricated with different lattice constants. We then demonstrate the ability to define the electron ensemble (the number of electrons in the many-body system of the array) and characterize the charge addition boundaries and resonant tunneling within the 2D array using in-plane gates. Finally, we measure the transport spectrum at elevated temperatures and reveal thermally activated Hubbard bands within the array.

### Tuning the hopping amplitude and long-range interactions

It is well known[28] that the nearest neighbor hopping $t$ is exponentially dependent on the lattice-constant $a$ while the long-range e-e Coulomb interactions are inversely proportional to $a$. Figure 2a, d, g shows STM images after patterning from three arrays, where we intentionally

increase the lattice constant $a_1 = 4.1 \pm 0.3 nm$ in the first array, to $a_2 = 6.6 \pm 0.3 nm$ in the second array, and to $a_3 = 10.7 \pm 0.3 nm$ in the third array. If we interpret each artificial lattice site as an 'impurity' atom in silicon, the lattice constant in the first, second, and the third arrays correspond to a bulk doping density of $\sim 1.5 \times 10^{19}/cm^3$, $\sim 3.5 \times 10^{18}/cm^3$, and $\sim 8 \times 10^{17}/cm^3$, spanning from above to below the critical density of a metal-insulator transition in phosphorus-doped bulk silicon. As will be described in the following, transport measurement and charge stability analysis confirm a transition from delocalized electrons displaying metallic behavior in the first array to localized electrons in the third array.

Figure 2b, e, h are the measured charge stability diagrams from the three arrays measured at T ~ 10 mK base temperature (electron temperature ~300 mK). In the first array, charge addition boundaries run more or less parallel throughout the entire gate-gate plane, resembling the charge stability diagram of a single metallic island in a single-electron transistor (SET). Comparing the charge stability diagrams of the first array with the second and third arrays, a key observation is that the charge stability boundaries in the third array are dominated by straight line segments of distinct negative slopes with bias triangles at avoided crossings; whereas on the gate-gate plane in the second array, the charge addition boundaries appear crossed only at negative gate voltages and evolve into smooth curves at positive gate voltages. This is further substantiated by comparing the differential conductance bias spectra of the first and second arrays (Fig. 2c, f) measured along the dashed lines in Fig. 2b, e, respectively. Here, the Coulomb diamonds in the first array (Fig. 2c) are dominated by well-defined diamond shapes that are closing at small biases and of similar diamond heights (charging energy), resembling the conventional Coulomb blockade behavior of a metallic island SET. However, despite the dominant metallic behavior in the gate-gate map, we do observe

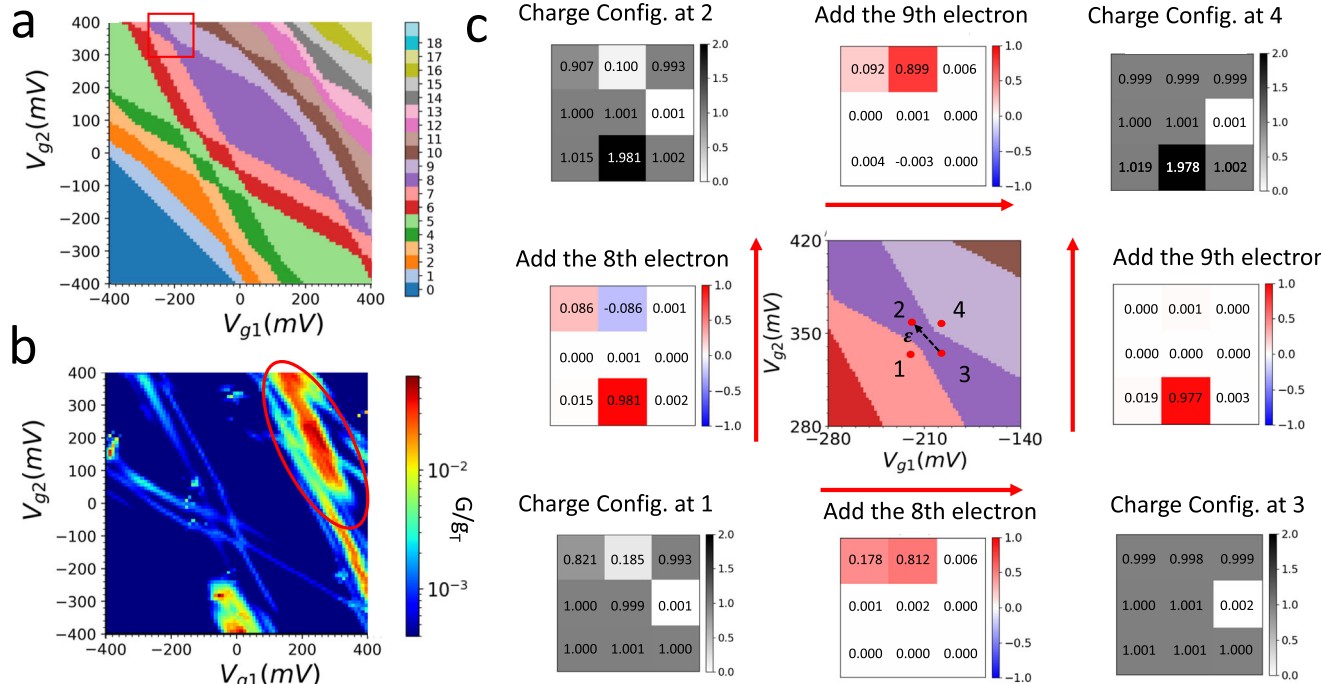

**Fig. 3 | Charge addition and resonant tunneling at avoided crossings in the simulated gate-gate map for Array #3. a** Numerically simulated charge stability diagram of the $3 \times 3$ array's ground state (see Methods). The hopping amplitude is set as $t = 0.5$ meV. **b** Numerically simulated resonant conductance gate-gate map (we account for the finite bias window in transport by equivalently setting $kT = 1$ meV in the simulation (see Methods for details). **c** Schematic illustration of eigenstate charge distributions and single charge addition at the charge stability diagram region highlighted by the red box in Fig. 3a. The charge configuration panels follow the dot numbering scheme from Fig. 1c, from left to right and from top to bottom, as D11, D12, D13, D21, D22, D23, D31, D32, D33. The black-white color maps represent the charge occupation of the ground states at select locations. The red-blue color maps represent the changes in ground state occupation when crossing a charge stability boundary and adding a single electron onto the array. The charge occupation numbers are overlaid on each charge configuration panel.

small discontinuities in the transport lines in the higher voltage range, likely due to nearby charge instabilities.

In contrast, the Coulomb diamonds of the second array (Fig. 2f) are irregularly shaped, and, despite the expected presence of disorder within the arrays, the bias spectrum of the second array features two groups of small Coulomb diamonds (highlighted by magenta color bars), corresponding to the upper and lower Hubbard bands, separated by a set of large, irregularly shaped Coulomb diamonds, corresponding to the finite analog of a Mott gap. The separation of the two Hubbard bands is characterized by the heights (addition energy) of the Coulomb diamonds at the Mott gap, ~50 meV, which is in reasonable agreement with the addition energy of individual quantum dots of few (1 to 3) dopant atoms[29]. (See Supplementary Note 3) Additionally, in Fig. 2c, f, we observe a qualitative difference in the bias conductance spectrum between the two arrays at base temperature. In the second array (Fig. 2f), conductance is visible as lines of increased differential conductance running parallel to the edges of Coulomb diamonds, indicating a discrete eigen-energy spectrum within the array; such a discrete conductance spectrum is not visible in the first array (Fig. 2c), indicating a quasi-continuous (metallic) density of state distribution in the first array, i.e., the exited addition energy state separation falls below $k_B T$ so the addition energy levels can no longer be individually distinguished at the operating temperature. (See Supplementary Note 4 for the impact of hopping, long-range interactions, and disorder on simulated addition energy spectrum and charge stability diagrams and Supplementary Note 5 for the effects from decreasing the lattice constants within the simulated array)[33].

**Gate-tuning the electron ensemble and charge distributions**
In this section, we use results from the third array, which is near the atomic-limit (weak tunnel coupling limit, $\frac{U}{t} \gg 1$) to illustrate tuning of the charge distributions within an array using in-plane gates. In Array

#3, since the source/drain leads are weakly tunnel coupled to the array and there is relatively large local and long-range electron-electron interactions, the source-drain conductance through the array is in the Coulomb blockade transport regime[34,35], as evidenced by the zero-conductance regions at finite bias in the measured conductance maps. The Coulomb blockade in Array #3 can be lifted by applying plunger gate voltages that overcome the interaction-induced blockade barrier and align an addition energy level of the array with the Fermi level in the source and drain leads. At these gate conditions the electron number in the array can fluctuate by one (finite compressibility), allowing source-drain conductance through the array.

Figure 3a, b plot numerically simulated charge stability diagrams of the ground states and conductance map over the gate-gate space for Array #3. Within each color domain in the simulated charge stability diagram, the total number of excess charges in the array ($N$) is constant, corresponding to the Coulomb blockaded regions in transport. Sweeping the common-voltage applied to both gates along the 45-degree diagonal direction on the charge stability diagram controls the total number of electrons, therefore, the filling factor of the model. Sweeping the differential voltage between the two in-plane gates (along the 135-degree diagonal direction in Fig. 3a) effectively tilts the chemical potential landscape within the array, altering the charge distribution of the ground state without considerably affecting the filling factor within the array. Due to variations in the input binding energies (see Supplementary Note 2), the largest Coulomb blockaded region in the theory diagram belongs to $N = 8$, corresponding to ~11% hole doping, as opposed to half filling ($N = 9$), which is expected for the uniform Fermi-Hubbard model.

The charge stability domain boundaries correspond to conductance lines in a transport map. Comparing the simulated conductance map with the charge stability diagram, not all charge addition boundaries are equally visible in conductance. This is also

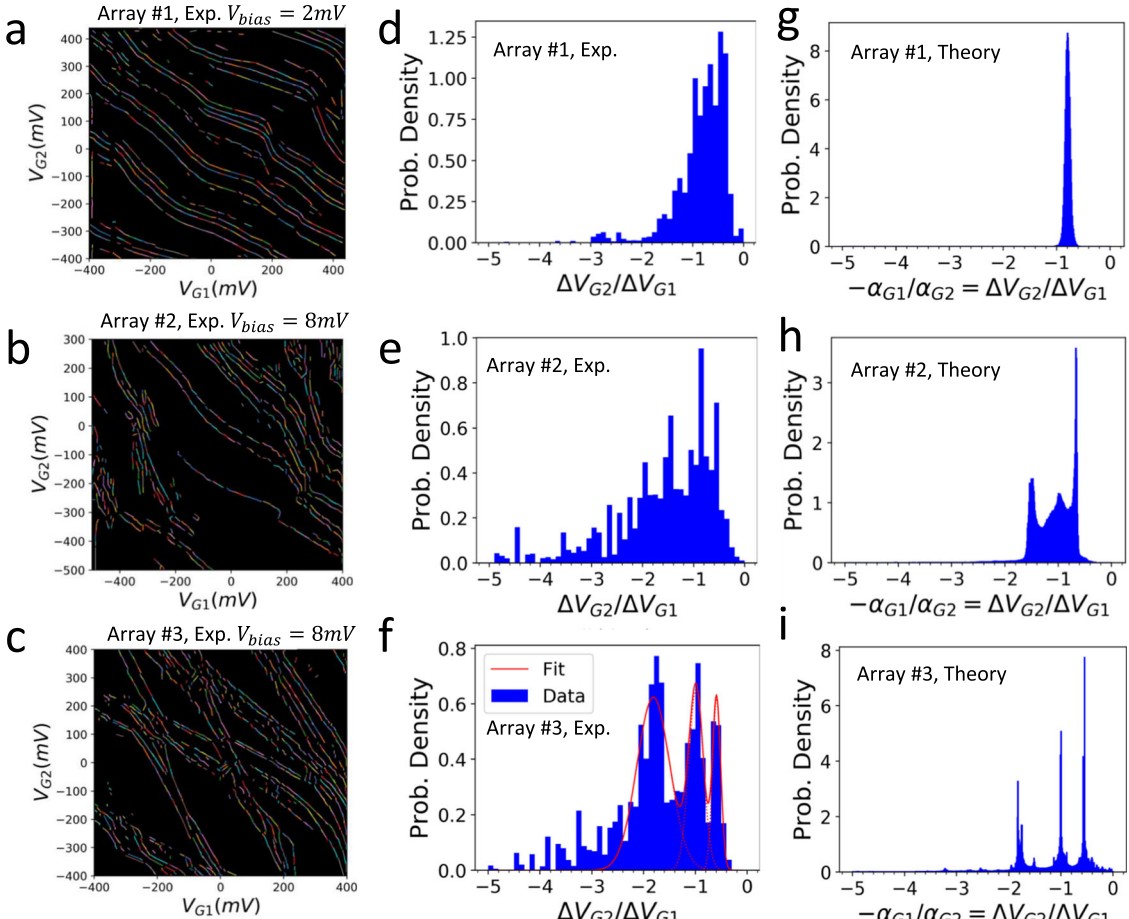

**Fig. 4 | Histogram distributions of conductance features in gate-gate maps for all three arrays.** The slopes of the conductance lines contain spatial information about the eigenstates through which addition electron transport occurs. **a–c** Edge detection of the conductance features in the measured gate-gate maps of the first array at $V_{bias} = 2mV$ (**a**), the second array at $V_{bias} = 8mV$ (**b**), and the third array at $V_{bias} = 8mV$ (**c**), respectively. The edge detection is performed using a Hough transform. The detected edges of constant slopes are represented by colored line segments. We quantify the span of each line segment by measuring the length of the line segment in units of pixel in the gate-gate map. **d–f** Histogram of the detected conductance line slopes $\triangle V_{G2}/\triangle V_{G1}$ of the first (**d**), the second (**e**), and the third (**f**) arrays, respectively. The histograms are plotted as probability density distributions of the number of pixels of the detected conductance line segments as a function of their slopes from the gate-gate maps. The three histogram peaks in (**f**) are fitted using Gaussian functions. **g–i** The calculated distributions of the gate lever arm ratio $-\alpha_{G1}/\alpha_{G2}$ of electron addition occupancies over the simulated charge stability maps of the first array (**g**), the second (**h**), and the third (**i**) arrays, respectively.

evident in the non-closing diamonds that are circled in Fig. 2i[36]. Both the measured and simulated conductance maps agree on a relatively large conductance along a particular line of negative slope in the positive gate voltages region of the diagram (circled in Figs. 2h and 3b). Visibility of the charge addition boundaries in measured conductance maps can be enhanced by increasing the bias window (see Supplementary Note 8) to allow transport through excited states or other inelastic/incoherent transport processes in the array mediated by, for example, electron-phonon interactions. We attribute the non-symmetric shape of the charge stability diagrams with respect to two gate voltages, or the fact that Coulomb diamonds do not always close (do not have conductance) at a small bias along the diagonal gate-gate direction, to the following: First, we expect the dominant effect to be the non-uniformity in binding energies due to variations in the number of dopants per site, which not only makes the system's energy landscape non-uniform for the electrons but also affects the transport through the array. Additional effects include small asymmetries in the way each gate affects individual site potentials and atomic-scale variation in the hopping amplitudes.

Effects of reduced lattice constants and increased hopping amplitudes for the first and the second arrays, where $t = 8$ meV and $t = 2$ meV, respectively, can be seen in simulated charge stability and conductance diagrams (see Supplementary Section S8 and Fig. S9). To illustrate the delocalization of many-body states in the second array, Fig. S9c plots ground state charge and charge addition distributions at select positions near an avoided crossing. Figure 3c shows ground state charge and charge addition distributions at selected positions near an avoided crossing at $N = 8$ in the simulated charge stability diagram in Fig. 3a. The simulations confirm that electron additions take place mostly locally at individual sites on this array. Comparisons to similar plots for the second array (see Supplementary Note 5) reveal the effect of reduced lattice constants and increased hopping amplitudes ($t = 2$ meV) in delocalizing the added electrons.

With our symmetric, in-plane two-gate design, the ratio of the upper gate's (Gate1) and lower gate's (Gate2) capacitive coupling to electrons at a quantum dot is determined by the ratio of the quantum dot's distances to the upper gate and to the lower gate. Charge additions to quantum dots in the upper, middle, and lower rows for Array #3 correspond to three distinct lever arm ratios, which manifest in the charge stability diagram as three different slopes in charge addition boundaries. For example, in Fig. 3c, when gating the array from positions 1 to 2 and from 3 to 4, we cross charge addition boundaries of the same slope, corresponding to the addition of an electron onto sites in the lower row. Similarly, when gating the array from positions 1 to 3

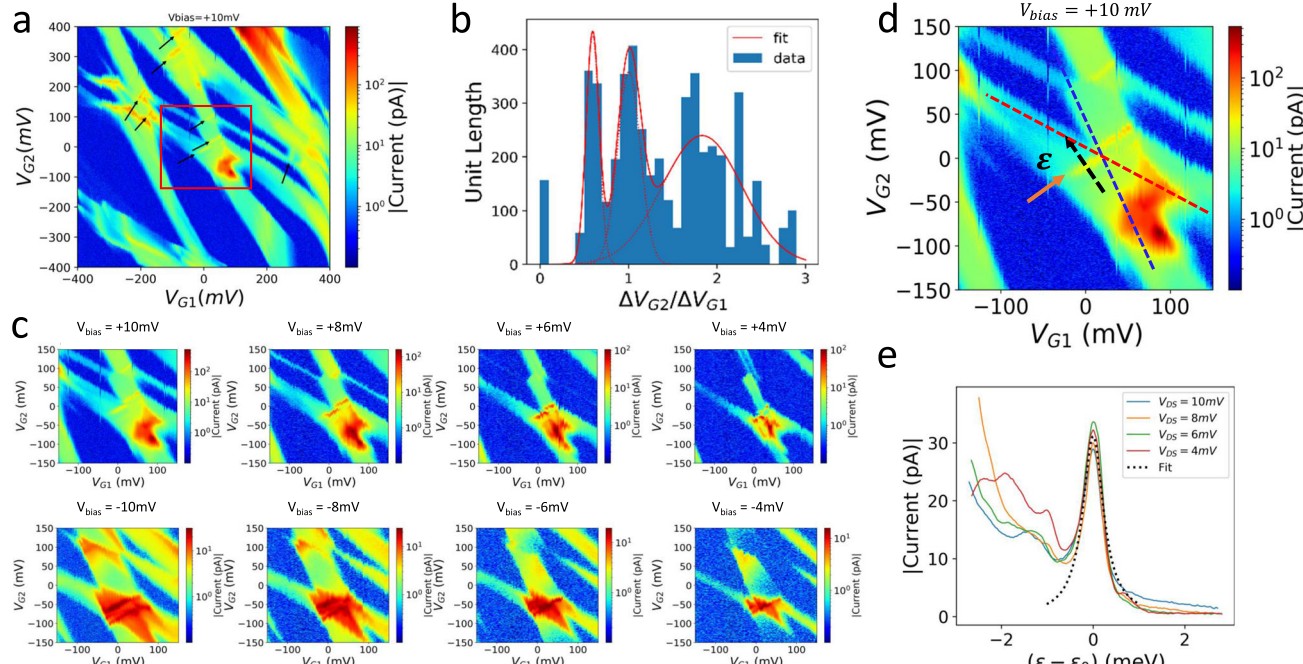

**Fig. 5 | Gate-gate maps showing positively sloped resonance conductance lines resulting from hybridization in many-body states in the multi-island limit (Array #3). a** The measured charge stability diagram at a bias window of +10 mV. Conductance resonance lines with positive slopes are marked by black errors. **b** Histogram of the positive slopes of conductance resonance lines extracted using edge-detection on the measured charge stability diagrams of the third array over a range of bias voltages from $V_{bias} = -10$ meV to +10 mV. The y-axis of the histogram is in arbitrary units of length of the extracted positive-slope conductance line segments. **c** Bias evolution of the resonant conductance at the avoided crossings that

are highlighted by the red box in (**a**). **d** Close-up of a region containing a few avoided crossings as highlighted by the red box in (**a**). The solid arrow indicates a resonant conductance line. The dashed lines mark the slopes of the charge addition conductance lines that come across near the resonant transitions. The dashed arrow marks the gate detuning $\varepsilon$ axis perpendicular to the resonant conductance line. **e** Overlaid resonant conductance profiles along the gate detuning axis that are measured at different biases. Each curve is fitted using Eq. 2 in the main text; and the best-fit parameters are averaged to reconstruct the dotted curves.

and from 2 to 4, the charge addition boundary slope corresponds to adding an electron onto sites in the upper row. Charge addition to the middle row can be found elsewhere in Fig. 3a when crossing addition boundaries with an intermediate magnitude in slope.

We carry out a quantitative analysis of slopes by running edge detection algorithms on the measured charge stability maps of the three arrays (See Fig. 4a–c) and plotting histograms of the detected edge length along their negative slopes $\frac{-\triangle V_{G2}}{\triangle V_{G1}}$. As can be seen in Fig. 4d, e, f, the histogram of the first array (Fig. 4d), shows a single peak in the distribution, mimicking the behavior of conductance through a metallic SET island. In the second array (Fig. 4e), in contrast, the single peak in the distribution evolves into a broad distribution, exhibiting conductance that occurs via single addition electrons that are more often shared simultaneously by sites across different rows. The third array shows three narrow distinct peaks, corresponding to charge addition sites in the upper, middle, and lower rows, with little mixing in states between different rows for conductance via single electrons jumping on and off the array. This change in slope distributions is characteristic of the transition from metallic to the collective Coulomb blockade regime upon decreased hopping amplitude relative to the interaction strength[34]. Such a transition from metallic behavior to the collective Coulomb blockade regime, where the energy gap inhibits adding an electron to an already strongly correlated state, is also observed in Fig. 2c, f as the many-body energy spectrum transitions from quasi-continuous to discrete distributions from the first to the second array. The extended skewing in slope distribution towards more negative slopes ($\triangle V_{G2}/\triangle V_{G1} < -3$) is likely the result of imperfect symmetry in device geometry and gate nonlinearity. Similar results are found in Fig. 4g, h, i plotting the distributions of the lever arm ratios of charge additions, which are calculated from the simulated charge

stability diagrams using the identical bias windows as in the experimental plots to include charge addition configurations of exited many-body states. (see Supplementary Note 1 for detailed capacitance and lever arm calculation methods). We note that the agreement in the shapes of the slope distributions remains qualitative due to unknown disorder and real measurement conditions that are not fully accounted for in theory.

### Resonant tunneling through many-body states

Additionally, we find small resonant tunneling lines with positive slopes in the conductance map from Array #3 (see Fig. 5a), which are attributed to hybridization between many-body states with the same filling factor but different charge distributions on the array. Because of the "multiply connected" topology[37] in the 2D array, the resonance conditions within the array enhance conductance through the array by opening additional conduction paths. Figure 5b shows a histogram of the positive slopes of conductance resonance lines extracted from the charge stability diagrams of Array #3 over a bias range from +10 mV to −10 mV. An example of the bias evolution of a positively sloped resonance conductance region is shown in Fig. 5c. We attribute the difference in bias triangles at positive and negative biases to disorder in the array. From the histogram in Fig. 5b, we observe positive slope distributions centering at three distinct values representing resonant tunneling between many-body states that are primarily localized at the first and the second rows, at the first and the third rows, and at the second and the third rows of the quantum dot array, respectively. Examples are ground states at positions 2 and 3 in Fig. 3c with an avoided crossing of charge addition lines between them. The slopes of the two crossing charge addition lines (dashed lines in Fig. 5d) correspond to charge addition to sites in the upper row and to sites in the

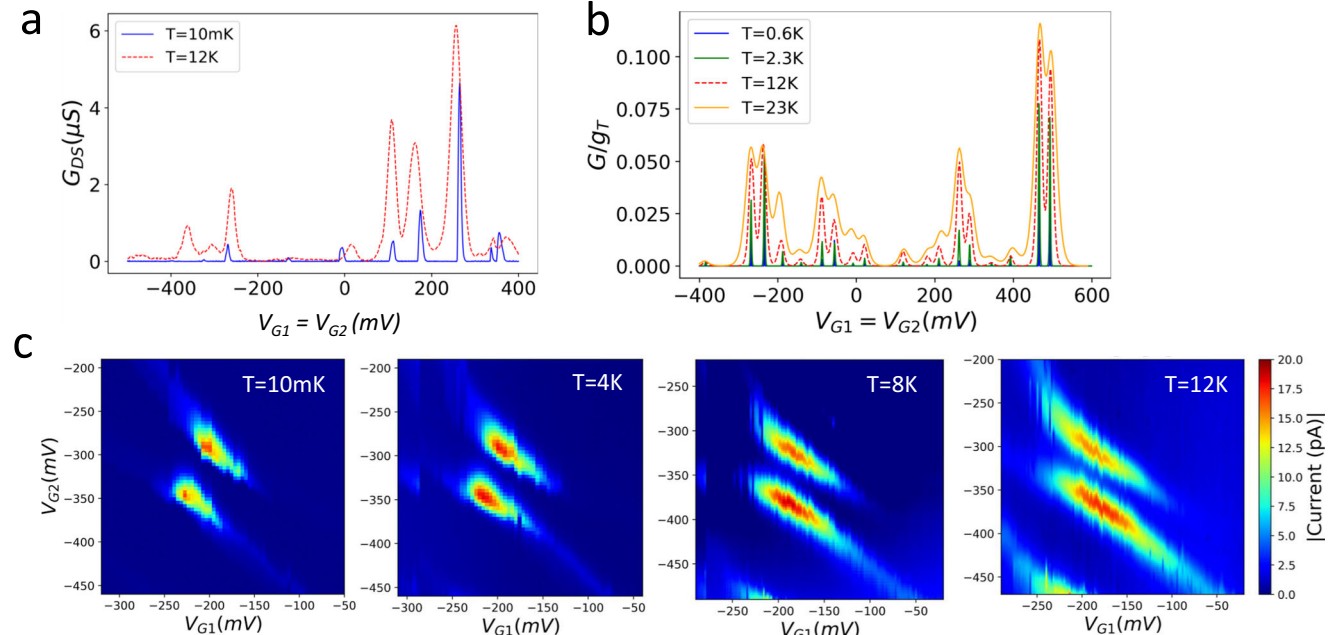

**Fig. 6 | Characterizing thermal activation in the second array at elevated temperatures. a** Direct current conductance measured along the dashed line in Fig. 2e at a bias $V_{bias} = 3mV$ and at temperatures $T = 10mK, 12K$. **b** Simulated Coulomb oscillation peaks in conductance of the second array at different temperatures (See Methods). **c** Conductance at an avoided crossing region of the second array (highlighted by the circle in Fig. 2e) that is measured with a bias $V_{bias} = 6mV$ and at varying temperatures $T = 10mK$, $4K$, $8K$, $12K$. All 4 plots in (**c**) share the same color bar on the right.

lower row, respectively. Approximating the two ground states as classical states of a double-dot system in which the electron is localized at either the left or the right dot, we can find an estimate for the effective tunneling between the states, t′, by fitting the conductance peak at the avoided crossing to the following Lorentzian form[38,39]

$$\frac{I_{ds}}{e} = \frac{t'^2 \Gamma_d}{t'^2 \left(2 + \frac{\Gamma_d}{\Gamma_s}\right) + \frac{\Gamma_d^2}{4} + \frac{1}{h^2}\left(\varepsilon - \varepsilon_0\right)^2} \tag{2}$$

where $I_{ds}$ is the drain-source current; $t'$ is the tunnel coupling rate ($1GHz \approx 4.14\mu eV$) between two charge occupations of discrete energy levels. $\Gamma_d$ and $\Gamma_s$ are tunneling rates to or from the drain and source leads, which are assumed to be approximately equal in this study; $(\varepsilon - \varepsilon_0)$ is the detuning energy between the two discrete energy levels (see black dashed arrow in Fig. 5d, based on $V_{G1}$ and $V_{G2}$ using lever arms, see Supplementary Note 1); and $h$ and $e$ are Plank's constant and the charge of a single electron. The amplitude of the current peak is primarily determined by $\Gamma_d$ and $\Gamma_s$; and the width of the peak depends mostly on $t'$. We treat $t'$, $\Gamma_d$, and $\Gamma_s$ as fitting parameters.

As shown in Fig. 5d, e, we perform the fit using the measured conductance near a crossing along the black dashed line in Fig. 5d. We find $t' \approx 4.4 \pm 0.3GHz (14.1 \pm 1.7\mu eV)$ and $\Gamma_d \approx \Gamma_s \approx 0.6 \pm 0.1GHz$. The insensitivity of the resonant current peak height and peak shape to the applied bias confirms that the observed peaks are dominated by resonant transport through two discrete energy levels. Away from zero-detuning, the two discrete energy levels are isolated, while at zero-detuning, the two discrete energy levels maximally hybridize. For a double-dot system, the separation at an avoided crossing is determined by $(U_m + 2t')$, where $U_m$ is the long-range e-e repulsion between the two sites. We computationally verify that this relationship also applies to our $3 \times 3$ array systems where two localized charge distribution configurations that are on resonance at an avoided crossing can be mapped to the two sites of a double-dot system. (See Supplementary Note 8) Here each $t'$ should be interpreted as an off-diagonal matrix element for an effective two-level Hamiltonian near the avoided

crossing and distinguished from the nearest neighbor hopping amplitude $t$ in the Fermi-Hubbard Hamiltonian. However, we find that the magnitude of $t'$ from the above fit is consistent with a simulated tunnel coupling[40] that is between non-neighboring array sites in the upper and lower rows.

## Thermally activated transport within an array

Transport through the second array is of special interest because its lattice site density corresponds to the critical doping density in silicon between tunneling and band transport regimes, where transitions of the electron system from a frozen Wigner-like phase to a Fermi glass has been previously observed by increasing the temperature in a few-dopant silicon transistor[41]. Here, we explore the thermal activation in the second array by monitoring the temperature evolution of transport properties through the array. The experimental temperature range investigated is limited by the temperature (~16 K) at which leakage current through the Si substrate becomes comparable to conductance through the array. At elevated temperatures, the charge occupation is thermally broadened not only in the Fermi distributions within the source/drain leads, but also in the occupation of the many-body states within the array. In addition to reducing the opacity at the source and drain tunnel junctions, both thermal effects increase the ability for addition electrons to access many-body excited states, and therefore, open new tunneling paths within the array that were previously not accessible at base temperature.

Figure 6a plots the Coulomb oscillation in source-drain conductance at T = 10 mK and 12 K taken along the dashed line cut in Fig. 2e using a small bias voltage of $3mV$. We observe the removal of the Coulomb blockade within each of the Hubbard bands at higher temperatures, where charge stability domains characterizing fixed numbers of electrons in the array become ill-defined except for at the large finite analog Mott gap. We also observe a similar transition in transport in the simulated Coulomb oscillations at elevated temperatures (Fig. 6b), which can be explained by a transition from the Coulomb blockade regime to the collective Coulomb blockade regime (two Hubbard bands)[34]. In this system, thermal broadening of

Coulomb oscillation peaks at T = 12 K becomes large enough to eliminate smaller Coulomb blockade gaps within the upper and lower Hubbard bands but not the large gap in between. Such thermally activated Hubbard band formation behavior within a 2D array is a key distinction from the thermal broadening in a zero-dimensional system where Hubbard bands do not exist."

Figure 6c shows the temperature evolution in conductance at an avoided crossing region of the second array (circled in Fig. 2e) at T = 10 mK, 4 K, 8 K, and 12 K. Conductance at the two triple-points at the avoided crossing stretches from well-defined bias triangles to conductance lines running in parallel, in analogy to the development of crescents at the triple-points of a double-dot system when increasing the tunnel-coupling between the dots[42]. Within the 2D array, this signature can be attributed to the additional hopping channels that become accessible at elevated temperatures (thermally activated hopping), that not only enhances conductance through the array, but also causes many-body states to become more delocalized within the array.

## Discussion

We have simulated an extended Fermi-Hubbard Hamiltonian of a 3 × 3 2D lattice using a series of STM-fabricated 3 × 3 arrays of few-dopant quantum dots and probed the many-body properties within the arrays using quantum transport measurements. By introducing two in-plane gates on either side of the array, we demonstrate gate tuning of the electron ensemble within the array as well as the flexibility of tilting the chemical potential landscape for characterizing the charge distributions of many-body ground states and resonant properties between them. By varying the lattice constants of a series of array devices at the fabrication stage, we demonstrate effective tuning of the hopping amplitude and long-range interactions within the simulated Hamiltonian. Through comparisons with theory, we can identify a finite-size analogue of the transition from metallic to Mott insulating behavior in charge distributions within the array when the lattice constant is increased from $4.1 \pm 0.3 nm$ to $10.7 \pm 0.3 nm$. By increasing the temperature near a critical lattice site density, we observe the formation of thermally activated Hubbard bands and enhanced electron transport through the array via thermally activated occupation.

In this first experimental realization of an extended Fermi-Hubbard Hamiltonian using atomically patterned 2D arrays of dopants, the quantum simulation accuracy is limited by uncertainties in the exact number of dopant atoms and their clustering configurations within each array site, as well as atomic-scale variations in nearest-neighbor hopping amplitudes. Continuing efforts are underway within our group and in the research community to seek long term solutions in improving the atomic perfection in dopant-based quantum devices. In a separate study[43], we recently demonstrated fabrication of a sample having 6 sites and 12 sites with precisely one single dopant per site by combining low-temperature feedback-controlled hydrogen lithography[44] with a new method called feedback-controlled manipulation. The precise number of dopant atoms in a few-dopant quantum dot and its clustering configuration can be measured using STM spectroscopy after dopant incorporation and before Si epitaxial overgrowth. When combined with ab-initio calculations of the electronic structure of STM-measured dopant-cluster configurations as inputs, we expect significant improvement in the agreement between the experiment and theory in the future. In addition, development of more complex virtual gate designs[7] is underway allowing control of the chemical potential at each array site and fine tuning of local tunnel coupling, improving the accuracy of dopant-based analog quantum simulators in the 2 by n configuration.

These experiments confirm the viability of simulating a Fermi-Hubbard model using dopant-based 2D arrays that account for limited nonuniformity and pave the way towards more complex and accurate analog quantum simulations of extended Fermi-Hubbard

Hamiltonians using dopant atoms. Tunability within the 2D array demonstrates the expected control of Coulomb interactions, hopping, filling factors, and temperature; this provides a pathway to explore previously inaccessible regions in the multi-dimensional condensed matter phase space. Extending this work to larger dopant-based arrays should allow the study of many-body localization and competing magnetic and charge order on 2D lattices, including frustrated geometries. In contrast to other quantum simulation platforms of Fermi-Hubbard Hamiltonians such as optical lattices, dopant-based artificial lattices are unique in reaching low effective temperatures and easy access to quantum transport. Additionally, relative to gate-defined quantum dot arrays, a dopant-based system has the unique property of having the naturally occurring nucleus at each dopant-based lattice site, allowing simulations that include nuclear spins and hyperfine interactions inherent in real-world condensed matter physics.

With continued advances to reduce disorder and improve atomic-scale precision in single-dopant placement, this method can be generalized to larger lattice arrays and other types of dopant species, such as boron[45] and arsenic[19], potentially embedded in alternative substrate lattice environments, such as a Ge substrate[46]. The results demonstrated in this study serve as a launching point for a new class of engineered artificial lattice systems to explore strongly correlated many-body systems in the solid state and explore less well-understood phenomena such as the pseudogap phase, strange metals, topological phases, and superconductivity in the Fermi-Hubbard model and magnetic ordering and frustration in spin systems. We point out that the problem of solving the Hubbard Hamiltonian for generic parameters quickly becomes intractable with state-of-the-art classical computers (even for ground state properties) for systems larger than 5 × 5 lattice sites. Even the 3 × 3 arrays in this study are on the cusp of what can be numerically done today exactly for finite-temperature transport properties.

## Methods

### Device fabrication and measurements

The 3 × 3 few-dopant quantum dot arrays are fabricated in an ultrahigh vacuum (UHV) environment with a base pressure below $4 \times 10^{-9}$ Pa ($3 \times 10^{-11}$ Torr) using an STM tip to create atomic-scale lithographic patterns of the device by removing individual hydrogen atoms on an hydrogen-terminated, Si(100) 2 × 1 reconstructed surface[44]. Detailed sample preparation, UHV sample cleaning, hydrogen-resist formation, and STM tip fabrication and cleaning procedures have been published elsewhere[47,48]. The substrate is then saturation-dosed with $PH_3$ gas at room temperature, where $PH_3$ molecules selectively absorb only onto the lithographic regions where chemically reactive Si dangling bonds are exposed. A subsequent rapid thermal anneal process at 350 °C for 1 min incorporates the absorbed phosphorus atoms into the Si surface lattice sites while preserving the hydrogen resist to confine dopants within the patterned regions. We embed the incorporated dopant atoms in a single crystalline silicon environment using low-temperature Si epitaxial overgrowth with an optimized locking layer to suppress dopant movement at the atomic scale during epitaxial overgrowth[26,48]. Finally, ohmic contacts to the buried device are formed using a low thermal budget contacting technique[27]. The number of incorporated P atoms in each quantum dot can be estimated by analyzing the lithographic sites that are available for P incorporation[31]. See Supplementary Note 2 for analyzing the best estimates of dopant numbers at each lattice site. Using the Si(100) 2 × 1 surface reconstruction lattice as an atomically precise ruler, the lattice constant of the fabricated arrays (center-to-center distance between adjacent quantum dots) can be precisely characterized. The STM-patterned in-plane source/drain leads and gates are saturation-doped[49] with a dopant density of $\sim 2 \times 10^{14}/cm^2$, corresponding to a 3D doping density of $\sim 2 \times 10^{21}/cm^3$ that is approximately three orders of

magnitude above the metal-insulator transition, allowing quasi-metallic conduction in all electrodes. We carry out direct-current (DC) transport measurement of the device inside a dilution refrigerator at a base temperature of ~10 mK.

## Numerical simulation

We have constructed a time-independent extended Hubbard model to simulate the $3 \times 3$ quantum dot array system following the analysis first introduced by ref. 50 for semiconductor quantum dots and more recently by Le and coworkers[28] for dopant arrays in silicon. In our extended Hubbard model (see Eq. 1), we include one spinful orbital per dot near the Fermi level, which means a maximum of 18 valence electrons can be added onto the array. We use a finite element software package, FasterCAP[51], for capacitance matrix calculations based on STM imaging of device geometry patterned using STM-lithography, where a lateral seam of 2.5 nm and a vertical thickness of 2 nm have been included to account for the Bohr-radius-like electron density extension beyond the lithographic patterns, previously shown to be necessary to reproduce the experimental capacitance values in STM-patterned Si:P devices[15,52]. We formulate the electrostatic potential landscape as well as on-site and long-range electron-electron Coulomb interactions using the calculated capacitance matrix (See Supplementary Note 1). We estimate the number of dopant atoms in the quantum dots by characterizing the binding energies of similar few-dopant quantum dots with a similar lithographic patch (See Supplementary Note 3). We approximate the screened on-site electron-ion Coulomb interaction in a quantum dot using the charge-neutral binding energy of the estimated dopant cluster[29]. We adopt a Fermi level of −80 meV with respect to Si conduction band edge in the saturation doped, in-plane source/drain, and gate electrodes[53,54]. We approximate the long-range electron-ion Coulomb attractions using a point charge approximation[28]. More details about model parameter extractions are provided in the Supplementary Notes. Exchange interactions and longer-range hopping terms are not implemented in our model. We assume the lead's tunnel coupling to the dot array is sufficiently weak so that the dot array can be treated as an isolated system when solving for the eigenstates.

At each gate-gate point, we use exact diagonalization to solve for the ground and excited many-body eigenstates of the Hamiltonian, whose matrix is represented in the Fock basis, for particle numbers ranging from 0 to 18 in the $3 \times 3$ array system. The charge stability diagram simulations utilize the classical capacitance coupling and best-estimated numbers of dopants per site based on the STM-patterned device geometries (See Supplementary Note 2). The hopping amplitudes are set as $t = 8$ meV, $t = 2$ meV, and $t = 0.5$ meV for simulating the first, second, and third arrays, respectively, based on the designed lattice constants in the arrays[40]. Charge stability diagrams and ground state charge distributions are obtained using the Lanczos algorithm after further block-diagonalizing the Hamiltonian matrices based on the total spin in the $z$ direction. The numerically simulated conductance results in the main text are linear response conductance through the array at finite temperatures and zero bias that is calculated following the formalism of ref. 28, which uses Fermi's golden rule for the tunneling rate to/from the leads and a generalization of the transport equations developed by Beenakker[55] for a single quantum dot. For that, we implement full re-orthogonalization of vectors in the Krylov space in our Lanczos routine after each iteration and seek convergence for 25 to ~40 low-lying states if the size of the Hilbert space for a particular spin and particle number sector is greater than 2000. For that, except in Fig. 6b where we have used full diagonalization to obtain exact results, this approximation generally underestimates the conductance values, especially at elevated temperatures (T > 1 K), but does not alter its qualitative behavior. For smaller sizes, we perform full diagonalization using Intel's math kernel library. The experimental conductance gate-gate maps were measured

at finite bias voltage of the order of few mV and the extent to which excited many-body states participate in transport is largely determined by the bias window, rather than thermal broadening. The calculated conductance does not take into account the effect of the environment, including the leads and the bias window, or other processes, such as phonons, which can further broaden the measured conductance peaks even at very low temperatures. Therefore, we approximate these effects and the participation of excited states in transport by setting a 1 meV thermal broadening in the numerical calculations of conductance gate-gate maps of Fig. 3b in the main text and Supplementary Fig. 5b in the Supplementary Notes.

Adopting a similar notation as ref. 28, we write the conductance as

$$G = g_T \sum_{n,m} \sum_{\alpha,\beta} \frac{M_{\alpha,\beta}^{(L),n,m} M_{\alpha,\beta}^{(R),n,m}}{M_{\alpha,\beta}^{(L),n,m} + M_{\alpha,\beta}^{(R),n,m}} \times P_\alpha^{n,m} \left[ 1 - f_{FD}\left( E_\alpha^{n,m} - E_\beta^{n-1,m-1} \right) \right]$$

Where $g_T = e^2 \Gamma / (\hbar kT)$,

$$P_\alpha^{n,m} = \frac{\exp\left[ -(1/kT) E_\alpha^{n,m} \right]}{\sum_{n,m,\alpha} \exp\left[ -(1/kT) E_\alpha^{n,m} \right]}$$

and

$$M_{\alpha,\beta}^{(L)n,m} = \sum_{j \in cL} |\langle \Psi_\alpha^{n,m} | c_{j\uparrow}^\dagger | \Psi_\beta^{n-1,m-1} \rangle|^2$$

where the last sum runs over sites on the left edge of the array, closest to the source lead. $M_{\alpha,\beta}^{(R),n,m}$ has a similar formula with a sum that runs over sites on the right edge of the array. $E_\alpha^{n,m}$ is the $\alpha^{th}$ eigenenergy of the many-body wavefunction, $\Psi_\alpha^{n,m}$ with $n$ particles and a total spin of $m$. $k$ is Boltzmann's constant, and $f_{FD}$ is the Fermi–Dirac distribution function. $\Gamma$ is the hopping amplitude for electrons to/from the leads. In our plots, we show results for the dimensionless quantity $G/g_T$. We use spin inversion symmetry to simplify our calculations by considering the transport of only spin-up electrons through the array and inserting an overall factor of two in $G$ (Certain commercial equipment, instruments, or materials are identified in this paper to foster understanding. Such identification does not imply recommendation or endorsement by the national institute of standards and technology, nor does it imply that the materials or equipment identified are necessarily the best available for the purpose.).

## Data availability

All relevant data are available upon request from the authors.

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

## Acknowledgements

We acknowledge helpful discussions with John Randall and James Owen as well as Emily Townsend, Neil Zimmerman, Alexey Gorshkov, and Maicol A. Ochoa. This research was funded in part by the Department of Energy Advanced Manufacturing Office Award Number DE-EE0008311 and by a National Institute of Standards and Technology (NIST) Innovations in Measurement Science award, "Atom-Based Devices: Single Atom Transistors to Solid State Quantum Computing." E.K. was supported by the National Science Foundation under Grant No. DMR- 1918572. This work was performed in part at the Center for Nanoscale Science and Technology NanoFab at NIST.

## Author contributions

X.W. performed the measurements, X.W. and R.S. analyzed the data, X.W., J.W., and P.N. fabricated the devices. X.W. and E.K. formulated the theoretical model and carried out the numerical simulations. X.W., R.S., E.K., F.F., and G.B. contributed to the theoretical interpretation of the data. X.W., A.R., R.K., F.F., and R.S. contributed to electrical measurements. X.W., E.K., and R.S. wrote the manuscript with comments from all co-authors. R.S. conceived and supervised the project.

## Competing interests

The authors declare no competing interests.
