## [Peer Review File · Nature Communications]

REVIEWER COMMENTS

Reviewer #1 (Remarks to the Author):

This is an impressive experiment on the nanoscale transport through a STM fabricated 2D array of phosphorous donors in silicon. The authors claimed that they observed the features of the extended Hubbard model in the conductance signal including the Hubbard bands and the Mott gap. The donor separation was adjusted to demonstrate an interesting transition from an insulating Coulomb blockade phase to a metallic phase. Despite the significant uncertainty/disorder in the system the authors made great efforts in the theoretical simulation of the result, but a good agreement was not possible due to the lack of knowledge of the number of donors at each site.

This work will be of great interest to scientists working in semiconductor-based quantum technology, and it may open an exciting research direction in quantum simulation with dopant arrays. However, there are a few points that need clarification to make sure that the claims are valid.

The main disadvantage of the dopant-based quantum dots studied in this work is the uncertainty in the number of donors at each site which can range from zero to six (more likely from one to three). This leads to a very large disorder in the binding energy (75% change from 1P to 2P, for example). A quantum simulator is usually a well characterized system where the physical parameters are known with sufficient accuracy. Only then it is possible to simulate an unsolved model of many-body physics. Is it possible to eliminate the donor number uncertainty in the future, at least in principle? If not, then dopant-based quantum dots are not a promising platform for quantum simulation. In my opinion this work is more of an experiment on Coulomb blockade transport than quantum simulation.

Due to the uncertainty mentioned above the agreement between theory and experiment is only qualitative. Therefore, I am not sure that some of the conductance measurement results can be attributed to electrons tunneling through the many-body states of an extended Hubbard model as the authors claimed. In fact, most of the observations can also be explained by the simple picture of electrons tunnelling through isolated dots with different binding energies, i.e., the classical coulomb blockade regime. For example, the two bands in Fig. 7a can very well be due to electrons tunnelling through isolated dots whose binding energies are clustered around two separated values (a bimodal distribution). This would give rise to an apparent gap but obviously it is not the Mott gap, nor are the bands the Hubbard bands as claimed. The only evidence of many-body physics in the paper is the conductance resonance line with a positive slope shown in Fig. 3b and 3c and discussed in Equation 2. This line is attributed to the hybridization of two spatially separated many-body states in the

array. However, the line is tiny on the conductance map and hence I am not convinced that it is a strong evidence of state hybridization.

A well-known source of disorder in this system is the oscillation of the tunnel coupling with the donor separation (Koiller et. al. PRL 88 027903), which can be significant even if the variation of the donor separation is only a few lattice sites. This may lead to electron localization as strong as that caused by the binding energy disorder discussed in the paper, but it is completely neglected in the analysis of the experimental results. Is there any justification for this?

In summary, this is a high-quality piece of work in quantum electronics/transport that will be of interest to a wide range of researchers in solid-state physics and quantum information. There is a somewhat unjustified optimism about the system's potential for quantum simulation. The results obtained in this work are already at the limit of what can be achieved with a phosphorous array in silicon due to the uncertainty in the donor number at each site. There should be more justification/clarification on why the authors believed that the transport is mediated by the many-body states of the extended Hubbard model instead of the trivial states of isolated dots.

Other minor issues:

1. How exactly was the capacitance matrix calculated? There is a mesh shown in Fig. S1, so it is probably calculated with the finite-element method, but there is no mention of what algorithm/software is used.
2. line 52: "such as cold atoms and optical lattice systems" --- cold atoms and optical lattice systems are the same.
3. line 216: "the study of Anderson localization, and especially its fate in the presence of strong interactions" --- its phase.

Reviewer #2 (Remarks to the Author):

Summary of claims

Single/few-atom quantum dots in 3x3 lattices with different lattice constants between ~4 nm and ~11 nm were fabricated. They were measured by low-temperature transport, tuning of the electron ensemble by in-plane gates. The authors claim a tuning of hopping amplitude in the three lattices, to

transition from finite size analog of a transition from Mott insulating to metallic behaviour in the array. The authors claim a new solid-state platform for the quantum simulation of the extended Fermi-Hubbard model of strongly correlated 2D systems. Such results are of broad interest to the readership of Nature Communications.

In gate-gate maps, there is a distinct evolution from bias triangles in the first array with largest lattice constant in lithography. There are two slopes, for left and right edges and avoided crossings due to hybridization between them. In a second array, with intermediate lattice constant in lithography, there is a mix of bias triangles that evolve into single-island behaviour as more electrons are attracted into the array. In a third array, with smallest lattice constant, the data shows the signature of collective coulomb blockade (a metallic island).

I find the results are potentially interesting for an audience of Nature Communications. But several scientific and organizational points need to be addressed before this article can be published in Nature Communications.

Major comments/questions

1. In quantum simulation we wish to simulate a Hamiltonian that we know, but, that Hamiltonian is not well defined if we don't know the number of atoms on each site and their exact separation. What are the long-term prospects if atomic disorder can never be eliminated? What is the impact of the valley degree of freedom on these prospects? What are the prospects if the atomic disorder can be eliminated? How can it be eliminated or reduced?

2. Alternative explanations: An key claim of the paper is that the system is two-dimensional (that more than one "row" of quantum dots is playing a role). Is there some direct experimental evidence that the system is 2D? In other words, that rows of quantum dots closest to the drain and source are playing an important role in these transport measurements? There are a lot of calculations, but an experimental argument based on data and reasoning about the system that excludes other explanations (not detailed calculations) would be more convincing. For example, related to the current Fig. 2, how was it ruled out that this is not just a L/R resonance of two dots interacting directly with the leads?

3. Alternative explanations: In the discussion of the temperature dependence, the authors write "In this system, thermal broadening of Coulomb oscillation peaks at $T=12\text{K}$ becomes large enough to eliminate smaller Coulomb blockade gaps within the upper and lower Hubbard bands but not the large Mott gap in between. Such thermally activated Hubbard band formation behavior within a 2D array is a key distinction from the thermal broadening in a zero-dimensional system where Hubbard bands do not exist". This comes before any actual description of any experimental features. The

order should be reversed so the measurements are described first. What the possible explanations? How were some ruled out? The presentation in the manuscript can be improved in many places by following this convention, eliminating duplicated explanations, and reducing the overall length.

4. The authors argue that there is a resonant enhancement of transport due to hybridization between two different localized charge configurations that coupled differently to the left and right gates (config. 2 and 3 on Fig. 3a), which open additional conduction paths and reference 36 is cited. Can the authors elaborate on this and provide a qualitative description of the conduction enhancement process?

5. Can the authors explain more about why there are crossings at lower occupation numbers in Fig 4b (below the mott gap), but not on the higher occupation numbers (above the mott gap)?

6. Do you see transport closer to zero bias, or just at several mV biases (Fig. 2a, 2b, and Fig. 4b, 4e)? Those are very high biases.

7. Why are the calculations are done at $T = 1$ K, given that the data was taken at base temperature?

8. There is a discussion about non-closing diamonds in Fig. 2c (line 263). It would be very helpful for the reader if the authors point out exactly which diamonds they are referring to, and where they predict the closings would be. As is, it is a bit vague, because we don't know exactly which diamonds the authors refer to. That also makes it hard to understand lines 273-277. Additional description would be helpful.

9. "The middle row" -> It is explained that the charge configurations involving the middle row, with an intermediate slope, can be seen in the calculations in Fig. 2d. Are these observed in the data? If not, why are they not observed in the data?

10. Consider re-arranging to present the data in the following order: metallic limit, then intermediate, then multi-island last, because the multi-island measurement has additional description that is distracting. The main message would come through first, in this different order.

11. In my opinion, some of the figures can be put in the supplementary info. For example, Fig 5 can go there and Fig 6 can go there. It is enough to summarize the results in main text.

12. It is claimed in lines 273-277 that the asymmetry of Coulomb diamonds and non closing of some of them in Fig2C, is mainly due to the variation in binding energy. To support this claim, is it possible to extract a rough estimate of the binding energy variation from the data in Fig2C?

13. It is stated that inside their quantum simulator that applying a common voltage on both in-plane gates can be used to sweep the array's global chemical potential (line 177-178). However, this is unlikely to be the case, because the source and drain will screen the potential and make it non-uniform. Please include a more nuanced discussion of this.

Minor comments/questions

1. "readily achievable zero-temperature limit" on line 53. Something is wrong with this statement, since zero temperature cannot be achieved.
2. One advantage of the STM fabrication technique used by the authors is that it can place dopant atoms with ~ 1 nm precision. This is true. It is stated that ion implantation techniques are limited in resolution to 30-50 nm, online 63. However, higher implantation accuracies are possible, there is some literature suggestion ~ 5 nm is possible. Please include a short but more nuanced discussion of this.
3. It is stated that the STM-based hydrogen lithography was proposed by some authors and extensively developed at UNSW, on line 68. I think it is more accurate to say that it was developed, and expanded into a technique enabling the placement of dopant atoms on a surface at UNSW, and covering those with a thin layer of silicon
4. The authors use the convention that a positively charged site is called D^+ , a neutral site is called D^0 , and a negative charged site is called D^- (for example, line 145-148). This is the convention for single donors in the literature. I am not aware that this is used to describe multi-donor quantum dots. I don't think this convention should be used here.
5. Missing citation to go with "We ignored Coulomb exchange and higher-order hopping terms in our numerical model" on line 170.
6. To my knowledge, Anderson localization (line 215) involves the localization of non-interacting electrons, while the Anderson/Mott localization describes the transition for interacting electrons.
7. Line 220-221: "changing the lattice constants within the array" -> "measuring arrays fabricated to have different lattice constants".
8. Line 235: "at a base temperature of $T \sim 10$ mK" -> What is the electron temperature?
9. The parallel and serial transport configurations discussed in lines 269-270 are not well explained at that point. Are these the transport paths shown in Fig. 3 and Fig. 4?
10. The sentence "indicating a quasi-continuous (metallic) density of state distribution in the third array" on line 407 deserves further clarification. Does this just mean that the excited state separation falls below $k_B T$ so the states can no longer be individually distinguished at the operating temperature?
11. The sentence on line 443 should be clarified, why is the word melting used? Melting seems to imply something thermal. But nothing thermal happens when two parameters in a Hamiltonian vary with respect to each other. Some energy scale needs to be compared to temperature.
12. The comment about the Wigner-like phase on line 457 is counterintuitive to me. Wigner localization refers to spontaneous localization of electrons due to electron-electron Coulomb repulsion. But in a system with attractive ions, there is an explicit orbital localization due to the electron-ion attractive. Please explain what this has to do with Wigner localization.

13. Please explain the source of asymmetry in the line shape for the data in Fig3C.

14. The authors present the calculated charge configurations in Fig. 3a and 5c. Do the authors have any experimental data (perhaps from nearby charge sensors, if any) to support the calculations?

Reviewer #3 (Remarks to the Author):

The paper describes the electrical transport study of a small array of single/few dopant atom quantum dots in silicon, where the number of dopants per dot was nominally 1-3 but varied randomly, with limits in dopant number set by the size of the 'hole' in the hydrogen resist mask used to fabricate the array. Numerical simulations of the data provide insight into the mechanisms responsible for the transport phenomena observed. The work is of interest to a broad readership, and is extremely important, as it demonstrates a clear advance towards studying many body physics using atomic-scale dopant-in-silicon structures, made using the STM based hydrogen lithography method. This field has been evolving for the last 20 years, and recently resulted in the demonstration of a 2-qubit qubit gate, but still has not revealed much new fundamental physics. The study here paves the way for that to change, demonstrating the quantum simulation of an extended 2D Fermi-Hubbard Hamiltonian. The work is impressive and thorough, although some of the discrepancies between experiment and simulation (that I would expect) are brushed over a little too easily, as noted below. If the relatively minor issues highlighted below are resolved, then I highly recommend that the paper is published.

In general: While there is generally reasonable qualitative agreement between theory and simulation, and the simulation is invaluable for explaining the fundamental physics at play, the agreement between experiment and simulation is far from perfect. For example, Fig. 2a and 2b have the same trends as Fig. 2e, but there are also significant differences. Likewise Fig. 6d-i. This seems perfectly reasonable to me, as the actual numbers of dopant in each dot in each array is an educated guess rather than a known value. However, this is not sufficiently highlighted in the paper. The obvious place to do this would be in the discussion section. I would like to see a paragraph discussing how you would expect the agreement between theory and experiment to improve (which I presume it would) as the number and position of the dopants in the array became more accurately controlled, with the ideal situation of 1 dopant in each dot at a perfectly precise location. How much more closely would you expect the data and theory to agree in that situation.

In general: The paper discusses in great detail the uncertainty in the number of dopants in a dot. However, was the position of a dopant within a dot considered? For example, if D12 in array 1 had two dopants in it, the dopants could be ~ 0.4 nm apart or ~ 2 nm apart. How might this difference affect the transport data, if at all?

Specific comments to be addressed:

Page 8, Fig. 2e: Why has a different voltage range been chosen compared with the experimental data? I suspect it is so theory and experiment look more similar, but it does not make sense to do this. Please correct this, or state in the text why this is not done. Also, the red ellipses added to Fig. 2b and 2e are almost invisible when printed. These should be made clearer.

Page 10, after line 304: For clarity, the authors should explain in a bit more detail the physical interpretation of Fig. 3a. For example, later in the paper it is mentioned that the charge distributions and single charge addition in Fig. 5c demonstrate greater delocalisation of electrons throughout the array. However, delocalisation (or lack of it), and how it is inferred from the figure, was not described when discussing Fig. 3a. It would be helpful for the reader if it was. The addition, it is hard to immediately appreciate what is happening during the addition of the 8th electron Charge Config. 1 to 2, and 9th electron Charge Config. 2 to 4. For example in the eigenstate charge distributions of Charge Config. 2, it's hard to tell if the charge occupation in D11 is different to D12, and if D12 and D23 are different. I assume so, as following the addition of the 9th electron D11 and D12 look identical to the surrounding arrays. It also looks like 1.5 electrons are being added for the 9th electron. In addition to the figure, could you describe the charge occupation in D11 and D12 (for example) in terms of percentages of an electron charge, or whatever is most appropriate?

Page 15, line 437 (discussion of Fig. 6): The statement "Unstructured distributions at higher slopes ($\Delta VG2/\Delta VG1 < -3$) are likely due to combined effects of defects and edge detection artifacts" is unconvincing. What defects are expected? No extra dopants should be incorporated at the individual dangling bond sites. I do not see any evidence of buried dopants in the STM image. If edge detection artifacts cause that much of an effect, can they not be reduced by more sophisticated data processing methods? Is it not a more reasonable argument to say that the distribution of the number of dopants in the array is significantly broader than you think? For example, wouldn't dots with zero dopants skew the transport data more significantly than the dots that have non-zero numbers of dopants. The comparison between experiment and theory for array 1 (Fig. 6d and 6g) is reasonably convincing, despite the noise and the narrowing of the distribution for the arrays 2 and 3 is also very crudely in agreement between experiment and theory. However, I would not describe the agreement for the latter case to be good, and this should be commented on. Again, there is just no way of knowing exactly how many dopants are in each dot in the experiment so I would not expect good agreement. The authors should either state this, or a plausible alternative.

Page 25, Fig. S2d: For D13 in array 3, why is the lower bound assigned as 1 dopant (Table S2)? This 'dot' appears to be made from 2 regions of isolated desorption, with each region consisting of an identical line of 4 dangling bond (DB) pairs. Either you assume that 4 DB pairs have the possibility of

resulting in zero dopants, in which case the lower bound is zero, else the assumption is that at least 1 dopant must be incorporated at 4 DB pairs, in which case there will be 2 dopants in total.

Minor amendments:

Page 4, Fig. 1a: As this schematic is earlier in the paper than discussions of adding electrons to the array, the reader might try to correlate the number of valance electrons (shown as spins) on the quantum dots with the number of dopants in the dots. However I assume that the number of electrons in the array has been arbitrarily assigned for the schematic. This should be clarified in the caption. I would have liked to have seen the electrodes labelled also (Source, Drain, Gate), but I leave this to the discretion of the authors as it might then look less visually appealing.

Page 8, line 235: You should state that it is array 1 that you are discussing. It is not entirely obvious from the ending of the previous section (although not a great surprise!).

Page 12, Fig.5, caption: For (c) it should be mentioned that it is array 2 that is being discussed.

Page 19, Line 582: The reference should be Ref. 27 not Ref. 24 and it is Le et al,.

Figure 25, Figs. S2c and S2d: Enlarge the arrays (which are the important part for the associated discussion) in the middle of the image, cropping the images significantly if necessary.

Page 26: The font size changes on line 708.

Figure 27, Figs. S3c and S3d: It is hard to read the text or see the arrows clearly when the paper is printed. This should be rectified.

REVIEWER COMMENTS

We thank the reviewers for their thoughtful and insightful comments and questions. We have very carefully considered the reviewers concerns and comments and through additional analysis, new experimental data, and simulations, we believe we are able to comprehensively address the reviewers' concerns. We have provided a detailed, albeit lengthy, point-by-point response to the reviewer's comments.

This revision is quite substantial and includes a significant re-arrangement of the presentation, as requested by Reviewer 2, with the addition of new sets of experimental data and simulations in the main text and supplemental material. We have refocused the paper to more clearly make a compelling case that both many body physics and quantum fluctuations are not only expected but essential to understanding these strongly correlated Hubbard simulators. We did this in part by emphasizing the experimental signatures that strongly suggest a correlated, many body interpretation. We believe the paper has benefitted substantially from these revisions.

Following suggestions by Reviewer 2, we have reorganized the results to present the strongly tunnel coupled metallic array first, then the intermediate array, followed by the more complex weakly tunnel coupled array last. This involved reversing the order in a few figures and moving the "Tuning the Hopping Amplitude and Long-range Interactions" section to before the "Gate-tuning the Electron Ensemble and Charge Distributions" section. Importantly, throughout the manuscript we now label Array1 as the strongly tunnel coupled metallic array, Array2 as the intermediate coupled array in the transition regime, and Array 3 is now the weakly coupled array near the atomic limit. We changed substantively Fig. 2 and Fig. 3 and added an entirely new Fig. 5 with new data. Original Figure 5 was moved to the supplemental section and new data were added to that figure. New data and new figures are described below in the response to reviewers.

Reviewer #1 (Remarks to the Author):

This is an impressive experiment on the nanoscale transport through a STM fabricated 2D array of phosphorous donors in silicon. The authors claimed that they observed the features of the extended Hubbard model in the conductance signal including the Hubbard bands and the Mott gap. The donor separation was adjusted to demonstrate an interesting transition from an insulating Coulomb blockade phase to a metallic phase. Despite the significant uncertainty/disorder in the system the authors made great efforts in the theoretical simulation of the result, but a good agreement was not possible due to the lack of knowledge of the number of donors at each site.

This work will be of great interest to scientists working in semiconductor-based quantum technology, and it may open an exciting research direction in quantum simulation with dopant arrays. However, there are a few points that need clarification to make sure that the claims are valid.

The main disadvantage of the dopant-based quantum dots studied in this work is the uncertainty in the number of donors at each site which can range from zero to six (more likely from one to three). This leads to a very large disorder in the binding energy (75% change from 1P to 2P, for example). A quantum simulator is usually a well characterized system where the physical parameters are known with sufficient accuracy. Only then it is possible to simulate an unsolved model of many-body physics. Is it possible to eliminate the donor number uncertainty in the future, at least in principle? If not, then dopant-based quantum dots are not a promising platform for quantum simulation. In my opinion this work is more of an experiment on Coulomb blockade transport than quantum simulation.

The reviewer has identified an important current limitation in this first demonstration of AQS using dopants. We also recognize this limitation and have identified three different possible solutions to address the challenge of disorder, each of which shows promise to improve atomic precision for analog simulation.

1) The best long-term solution to improve the specified number of dopants incorporated per site is accomplished using feedback controlled, low temperature (LT) STM patterning methods and feedback-controlled dopant incorporation methods. Although all of the STM lithography used to make the devices for this manuscript was performed at room temperature, we have demonstrated in one of our recent articles, Nano letters 18 (12), 7502-7508, true atomic precision H-lithography at low temperature. Subsequently, we have combined LT feedback-controlled lithography with a new method called feedback-controlled manipulation. In recent results we have demonstrated 6 out of 6 lithography sites and 12 out of 12 sites to have incorporated only a single dopant per site and is a central outcome in a recent manuscript submission from our group (arXiv:2112.12200). This is the best, current long-term solution for definitive control the number of incorporated dopant atoms per site.

2) We can use the STM to measure the number of dopant atoms incorporated per site. Here, STM spectroscopy is used to identify the specific number of dopant atoms per site after phosphorus incorporation. While this does not improve the atomic precision of fabrication it improves the model input accuracy to achieve better theory to experiment comparisons (arXiv:2112.12200).

3) We are also developing fully gated devices for AQS. These are suitable for one dimensional arrays and $2 \times n$ arrays where the chemical potential and electron occupation can be controlled at each site. See the recent results from Simmons' group at the APS March meeting 2022 (<https://meetings.aps.org/Meeting/MAR22/Session/Z39.9>) as well as from our group (<https://meetings.aps.org/Meeting/MAR22/Session/B37.1>).

The reviewer has also correctly pointed out that, as shown in Table S3 in our supplementary material, variation in the number of dopant atoms per site leads to variations in the binding energy, and therefore, disorder in the chemical potential landscape of the array. On the other hand, the charging energy for the charge-neutral to charge-minus transition (see Supplementary Materials Table S3 last column) is only moderately dependent on the number of dopants per site in the one to three dopants range.

Due to the uncertainty mentioned above the agreement between theory and experiment is only qualitative. Therefore, I am not sure that some of the conductance measurement results can be

attributed to electrons tunneling through the many-body states of an extended Hubbard model as the authors claimed. In fact, most of the observations can also be explained by the simple picture of electrons tunnelling through isolated dots with different binding energies, i.e., the classical coulomb blockade regime.

The reviewer has asked an important fundamental question regarding the underlying physics and physical mechanisms of transport in our arrays. There are several arguments that support the many-body quantum interpretation and to be comprehensive, the answer is somewhat lengthy. In short, we believe the data, model, and the physics clearly support that many-body quantum effects are fundamental to this system and that electron-electron interactions and tunneling are intrinsic phenomena in this system and play an essential role in the array behavior with quantum fluctuations becoming dominant at reduced array site separations. The interaction effects become stronger as the array sites become closer, and the tunneling effects become weaker as the array sites become farther apart. New simulations in the supplemental material section S4 demonstrate that long-range interactions across the array sites play an important role and our measurements are not consistent with those for single or isolated quantum dots. Furthermore, with the increased t/U ratio in the first array (spacing as little as 4.1 nm), electron wavefunctions are certainly delocalized across the array and the individual nature of each lattice site is lost. While limits of no electron-electron interaction (the tight-binding model) or no tunnel coupling and interactions between array sites (the limit of many-body physics with quantum fluctuations, also known as the atomic limit) each lead to trivial scenarios, evidence shows we are not close to either limit and are, therefore, dealing with both many-body physics and quantum fluctuations in the arrays. We have added new material in the manuscript to more clearly discuss these effects, see page 7 in the revised manuscript for a discussion of the parameter values in this study in the context of relevant energy scales and the competing effects of long-range Coulomb interactions and hopping.

A key theme strengthened in the revised manuscript intended to make the quantum many body physics most evident is that by increasing the dot-to-dot separations from the first array to the third array, we explicitly elucidate the effects of tunnel coupling and electron-electron many-body interactions within the array. In the limit where the hopping/interaction ratio (t/U) approaches zero, “the simple picture of electrons tunneling through isolated dots” becomes a closer approximation, albeit in the presence of significant long-range electron-electron interactions. In the first array, the lattice sites are so close to each other that the electron wavefunctions delocalize across the array, and the individual nature of each lattice site becomes indistinguishable. This is borne out in gate-gate maps from the first array (experiment and simulation) where the charge transition lines do not have the distinct different slopes observed in Array 3 – this is direct experimental evidence that tunneling is becoming dominant, the transition from straight charge addition lines to charge addition lines that have distinct different slopes results from site specific capacitance coupling whereas in the delocalized quantum case the capacitance coupling is to addition electrons spread across the array. Another point of support of the many body physics is that there are many examples, such as the charge addition configuration examples of Fig. 3c in the main text and Fig. S5c in the supplementary materials, where the tunneling of single electrons onto or off the array or a single electron tunneling within the array involves modification of occupancies at more than two sites (multi-site participation) in a single tunneling event.

We also considered suggestions of the reviewer that the results are due to individual quantum dots that are tunnel-coupled directly to the source/drain leads. In this study, the source and drain tunnel junction gaps are ~ 7.8 nm for all three arrays and the array sizes are ~ 33 nm, 21 nm, and 12 nm for Array #3, #2, #1 respectively, following the new array naming convention in the revised manuscript. If only an isolated single dot is the transport mechanism, then the single electron has to tunnel through a tunnel barrier of least 24 nm in Array #3 (at least 18 nm in Array #2 and 14 nm in Array #1), the separation between the isolated dot and one of the leads. Based on theoretical [Gamble, J. K. et al. Phys. Rev. B 91, 235318 (2015), Le, N. et al., Phys. Rev. B 96, 245406 (2017)] and experimental [Wang, X., et al., Communications Physics, 3(1), 1-9 (2020)] results, we expect the tunneling conductance for sequential tunneling of an electron from the source electrode to an isolated dot to the drain electrode to be many orders of magnitude smaller than the observations in this study. Our 2020 Communications Physics manuscript presented a detailed investigation of transport in single electron transistors (SETs) as a function of separation between the quantum dot and source/drain leads. In a single quantum dot SET, a measurable current only flows when the tunnel separation is ~ 12 nm or less between source/drain and the quantum dot. 24 nm separations yield $\gg 10^9$ Ohm tunnel junction resistance and no measurable current. This has been validated in our double dot devices. [<https://meetings.aps.org/Meeting/MAR21/Session/E29.2>] Therefore, we have determined that the source-drain conductance mediated by a single isolated quantum dot does not support the observed level of conductance. Hence, the fact that we observe current is indicative of significant electron hopping between sites of the array, even in the weakly tunnel coupled array #3, which necessarily means quantum fluctuations, long range Coulomb interactions, and eigenstates of the many-body system are involved.

In our system, we do not have individual dot control in the 3x3 array and the potential landscape across the array has significant variation across the sites as a result of disorder, electron interactions, and electrostatic effects from the leads and gates. In Figure R1 below we show an example of the chemical potential across the array when adding the first electron to Array #3 and Array #2. This is a very different picture compared to a typical double dot experiment where each dot's electrostatic potential is individually controlled by gates allowing precise alignment of the chemical potentials and opening up a transport path at low source/drain bias voltage. In our system, these pathways open up as a direct result of many-body effects (including substantial local and long-range Coulomb interactions and their competition with hopping processes) determining the distribution and the extent of delocalization of electrons as they are added to the array. The formula for G in turn captures that by examining whether simultaneous charge variations on sites on the left and right edges are energetically allowed.

Figure R1. Simulated chemical potential map to add the first electron to (a) Array #3 and (b) Array #2 (following the new Array naming convention in the revised manuscript). The chemical potential energy at a site corresponds to the energy required to add the first electron onto the array and the electron is 100% localized at the site (see the H_μ term in Equations 1 in the main text). The chemical potential maps shown here are calculated when all the electrodes are grounded.

For example, the two bands in Fig. 7a can very well be due to electrons tunnelling through isolated dots whose binding energies are clustered around two separated values (a bimodal distribution). This would give rise to an apparent gap but obviously it is not the Mott gap, nor are the bands the Hubbard bands as claimed.

The binding energy and the charging energy at each quantum dot have different manifestations on a Coulomb oscillation measurement, as shown in Figure 7a (Figure 6a in the revised version). The width of a Coulomb blockade gap on the gate-voltage axis corresponds to the charge addition energy (scaled by gate lever arms) to add an electron onto the array system, arising from both the on-site and long-range electron-electron interaction terms. Based on our response to the previous comment above, we have excluded the possibility that the experimentally observed conductance arises from electrons tunneling through isolated quantum dots. Many-body and tunneling effects within the array system are needed to overcome disorder-induced localization and to mediate observable electron transport between the source and drain leads. Therefore, the observed transport spectrum through the array corresponds to the binding and addition energy spectrum of the many-body eigenstates within the array. Please see the added plots and discussions in the supplementary materials section S4 where we compare the simulated charge stability diagrams of Array #3 with and without interactions. This addition energy reduces to the single-particle energy spectrum in the noninteracting regime; however, in absence of interactions, the energy spectrum fails to exhibit the bimodal distribution as proposed by the reviewer.

Quantitatively, from Table S3 in the supplementary information, the on-site charging energy U is approximately the same ($\sim 45\text{meV}$) for isolated dots with 1 to 3 dopant atoms. The experimental gating range ($V_{G1,G2} \in [-400, 400]\text{mV}$ with the lever arm at a single quantum dot i on the order of $\alpha_{G1,i} + \alpha_{G2,i} \approx 0.2$) is large enough to overcome the on-site charging energies of single/few dopant quantum dots, and the on-site interactions play an important role in the measured energy spectrum of the array. The agreement between the apparent middle gap size and the on-site interaction U_i strongly points to its finite analog Mott gap nature. The smaller

energy gaps on either side, which smear out as the temperature increases, reflect the broadening of the addition energy spectrum due to other many-body effects, such as inter-site Coulomb interactions, and hopping.

Additionally, we added a discussion regarding energy scales on page 7 line 193 to clarify the many-body and quantum tunneling nature of the electron system in the arrays.

“Our best estimates point to an average ratio U_i/t that varies roughly from 6 to 90 from the first to the third array. The ratio of the nearest-neighbor Coulomb repulsion and the hopping amplitude also varies by an order of magnitude roughly from 2 to 20. So, while we expect Coulomb interactions to dominate in the third array, their strengths are comparable to, or less than, the noninteracting bandwidth (see SM, Figure S4(a)) in the metallic array, and therefore, we expect the significant tunneling/delocalization of electrons to change the character of the latter system. Disorder also plays a significant role in all three arrays. Apart from introducing site-to-site variations in interactions and tunneling amplitudes, the ratio of its strength (defined as the difference in binding energies for 1 and 3 dopants per site) and the hopping amplitudes varies roughly from 4 to 70 from the first to the third array.”

The only evidence of many-body physics in the paper is the conductance resonance line with a positive slope shown in Fig. 3b and 3c and discussed in Equation 2. This line is attributed to the hybridization of two spatially separated many-body states in the array. However, the line is tiny on the conductance map and hence I am not convinced that it is a strong evidence of state hybridization.

The conductance resonance line with a positive slope shown in Fig. 3b and 3c in the original version is only a representative example of the many positively sloped resonance lines scattered across the gate-gate conductance maps of the first array. In Fig. R2a, we use black arrows to highlight several occurrences of positively sloped resonant conductance lines at avoided crossings where hybridization of many-body states occurs. (Please note we have added this as new Figure 5 in the revised manuscript.) As illustrated in Fig. R2b, we observe slope distributions centering at three distinct positive values, corresponding to resonant charge oscillations and state hybridization between the first and the second rows, between the first and the third rows, and between the second and the third rows, respectively (here we follow the naming convention in the original manuscript and define the first (third) row as the row of quantum dots that is closest to Gate1 (Gate2), and the second row is in between the first and the third row). Due to the relative scarcity of the positively sloped resonance lines compared with the negatively sloped conductance lines over a gate-gate map, the statistical analysis in Fig. R2b includes edge detection data from gate-gate maps of a range of bias voltages from $V_{\text{bias}} = -10\text{mV}$ to $+10\text{mV}$. As an illustration in the evolution of resonant conductance over the range of bias voltages, Fig. R2(c) plots the positive slopes in the same region of Fig. 3b that are measured at different bias voltages. We have included Fig. R2 in the revised version of the main text.

Figure R2. Transport measurements of the gate-gate maps of source-drain conductance and the positively sloped resonance conductance lines of the first array. (a) The measured charge stability diagram at a bias of +10mV. Conductance resonance lines with positive slopes are marked by black errors. (b) Histogram distribution of the positive slopes of conductance resonance lines that are extracted using edge-detection from the measured charge stability diagrams of the first array over a range of bias voltages from $V_{bias} = -10$ meV to +10mV. (c) Bias evolution of the resonance conductance at the avoided crossings that are highlighted by the red box in (a).

A well-known source of disorder in this system is the oscillation of the tunnel coupling with the donor separation (Koiller et. al. PRL 88 027903), which can be significant even if the variation of the donor separation is only a few lattice sites. This may lead to electron localization as strong as that caused by the binding energy disorder discussed in the paper, but it is completely neglected in the analysis of the experimental results. Is there any justification for this?

As the review correctly points out, the tunnel coupling between two single dopant atoms oscillates sharply with dopant separations due to intervalley interference of the donor electron's wavefunction. The clustering of few dopants at each array site significantly affects the electron wavefunction and distribution as well as the oscillation in tunnel coupling. It has been shown that moving the dopants off the rectilinear axes (the arrays in this study orient in the [110]

directions) significantly reduces this oscillation for single dopants and with further reduction due to multiple dopants at each site. [Gamble, John King, et al. "Multivalley effective mass theory simulation of donors in silicon." *Physical Review B* 91.23 (2015): 235318.] In this study, since the exact number of incorporated dopant atoms and the exact dopant clustering configurations at each array site are unknown, we are unable to predict the exact disorder/variation configurations of tunnel coupling within the array. In addition, theoretical simulations have shown significant reduction of tunnel coupling oscillation between few-dopant quantum dots (private conversations with Eran Ginossar and coworkers). Therefore, we have not included the disorder in tunnel coupling in the analysis of the experimental result in this study.

In summary, this is a high-quality piece of work in quantum electronics/transport that will be of interest to a wide range of researchers in solid-state physics and quantum information. There is a somewhat unjustified optimism about the system's potential for quantum simulation. The results obtained in this work are already at the limit of what can be achieved with a phosphorous array in silicon due to the uncertainty in the donor number at each site. There should be more justification/clarification on why the authors believed that the transport is mediated by the many-body states of the extended Hubbard model instead of the trivial states of isolated dots.

See detailed justifications/clarification in response to the reviewer's previous comments above.

Other minor issues:

1. How exactly was the capacitance matrix calculated? There is a mesh shown in Fig. S1, so it is probably calculated with the finite-element method, but there is no mention of what algorithm/software is used.

We use a commercially available finite element software package, FasterCAP for all the capacitance matrix calculations in this study. We added these details in the Methods – Numerical simulation section in the main text.

Previous wording:

"We formulate the electrostatic potential landscape as well as on-site and long-range electron-electron Coulomb interactions using capacitance matrix calculations based on the STM-lithography patterns of the device geometry, where a lateral seam of 2.5 nm and a vertical thickness of 2 nm have been included to account for the Bohr-radius-like electron density extension beyond the lithographic patterns, shown previously to be necessary to reproduce the experimental capacitance values in STM-patterned Si:P devices.^{1,2}"

New wording:

"We use a finite element software package, FasterCAP, for capacitance matrix calculations based on the STM-lithography patterns of the device geometry, where a lateral seam of 2.5 nm and a vertical thickness of 2 nm have been included to account for the Bohr-radius-like electron density extension beyond the lithographic patterns, shown previously to be necessary to reproduce the experimental capacitance values in STM-patterned Si:P devices.^{1,2} We formulate the electrostatic potential landscape as well as on-site and long-range electron-electron Coulomb interactions using the calculated capacitance matrix."

2. line 52: “such as cold atoms and optical lattice systems” --- cold atoms and optical lattice systems are the same.

We have modified the description.

“Such as cold atoms in optical lattices.”

3. line 216: “the study of Anderson localization, and especially its fate in the presence of strong interactions” --- its phase.

We have changed the wording to,

“the study of Anderson localization, and especially its existence in the presence of strong interactions”

Reviewer #2 (Remarks to the Author):

Summary of claims

Single/few-atom quantum dots in 3x3 lattices with different lattice constants between ~4 nm and ~11 nm were fabricated. They were measured by low-temperature transport, tuning of the electron ensemble by in-plane gates. The authors claim a tuning of hopping amplitude in the three lattices, to transition from finite size analog of a transition from Mott insulating to metallic behaviour in the array. The authors claim a new solid-state platform for the quantum simulation of the extended Fermi-Hubbard model of strongly correlated 2D systems. Such results are of broad interest to the readership of Nature Communications.

In gate-gate maps, there is a distinct evolution from bias triangles in the first array with largest lattice constant in lithography. There are two slopes, for left and right edges and avoided crossings due to hybridization between them. In a second array, with intermediate lattice constant in lithography, there is a mix of bias triangles that evolve into single-island behaviour as more electrons are attracted into the array. In a third array, with smallest lattice constant, the data shows the signature of collective coulomb blockade (a metallic island).

I find the results are potentially interesting for an audience of Nature Communications. But several scientific and organizational points need to be addressed before this article can be published in Nature Communications.

Major comments/questions

1. In quantum simulation we wish to simulate a Hamiltonian that we know, but, that Hamiltonian is not well defined if we don't know the number of atoms on each site and their exact separation. What are the long-term prospects if atomic disorder can never be eliminated? What is the impact of the valley degree of freedom on these prospects? What are the prospects if the atomic disorder can be eliminated? How can it be eliminated or reduced?

The reviewer has identified an important current limitation in this first demonstration of AQS using dopants. We also recognize this limitation and have identified three different possible solutions to address the challenge of disorder. Please see the response to Reviewer 1, who has asked a very similar question (first response comment with a three-point response).

With respect to the impact of the valley degree of freedom on the above mentioned prospects, the presence of multiple conduction band valleys in silicon introduces additional complications to the dopant-based quantum simulation in silicon. The effects of valley splitting are not included in the Hubbard model and these effects have not been evaluated. In principle, if atomic disorder can be eliminated, dopant-based quantum simulators could enable the analog quantum simulation of valley physics in silicon quantum electronics.

2. Alternative explanations: A key claim of the paper is that the system is two-dimensional (that more than one “row” of quantum dots is playing a role). Is there some direct experimental evidence that the system is 2D? In other words, that rows of quantum dots closest to the drain and source are playing an important role in these transport measurements?

There are a few direct indicators of two-dimensional array behavior. First, please note that the source/drain separation to the nearest rows of quantum dots is chosen to be 10 dimer rows for all arrays and that these rows are sufficiently isolated from the source and drain electrodes to allow Coulomb blockade transport between the 3x3 array and the source/drain leads (the source and drain tunnel resistance to the array is much greater than the resistance quantum $\gg 26\text{k}\Omega$). Also, quantum dots (dots 21, 22, 23) in the middle column alone do not support the observed level of conductance due to their large separations from the source/drain leads. Here, we are using the naming convention from the paper such that the rows of quantum dots are parallel to and “connect” the source and drain leads while a column of quantum dots are in the direction of the gates and parallel to the edge of the source and drain leads.

The first indicator is that Array 3 has a distribution of three lever arms in the gate-gate map, each lever arm is associated with specific capacitive coupling to a row of sites. Each transport line in the gate-gate map is associated with a capacitive coupling lever arm to a row of sites whose electron number degeneracy supports transport. However, several occurrences of positively sloped resonant conductance lines at avoided crossings, where hybridization of states occurs, are observed. As illustrated in Fig. R3b, we observe slope distributions centering at three distinct positive values, corresponding to resonant charge oscillations and state hybridization between the first and the second rows, between the first and the third rows, and between the second and the third rows, respectively. This hybridization results in transport amongst dopants in different rows.

Additional direct experimental evidence is seen when comparing the gate-gate maps amongst the arrays, shown in Fig. 2b, 2e, and 2h in the main text, where the dot-to-dot tunnel barriers transition from relatively transparent (4.1 nm) to more opaque (10.7 nm) and the array properties transition from metallic to more localized electron states. With the increased t/U ratios in the first and second arrays, the quantum fluctuations become dominant. In the first array, the lattice sites are so close to each other that the electron wavefunctions delocalize across multiple rows in the array, and the individual nature of each lattice site becomes indistinguishable. This is borne out in gate-gate maps from the first array (experiment and

simulation) where the charge transition lines do not have the distinct different slopes as compared to Array 3. The transition from straight charge addition lines to those with distinct different slopes, as a result of changing long range electron-electron interactions and electron localization, is experimental evidence of both the two dimensional and quantum nature of the system.

There are a lot of calculations, but an experimental argument based on data and reasoning about the system that excludes other explanations (not detailed calculations) would be more convincing. For example, related to the current Fig. 2, how was it ruled out that this is not just a L/R resonance of two dots interacting directly with the leads?

The following are physical arguments that support the many-body transport picture. We have considered suggestions of the reviewer that the results are due to either an individual or several isolated quantum dots coupled directly to the source/drain leads and have determined that this does not support transport because the tunnel coupling is too weak. If only a single dot is the transport mechanism, then the resulting geometry gives rise to separations of up to 25 nm between the dot and one of the leads. Our 2019 Communications Physics manuscript presented a detailed investigation of transport in single electron transistors (SETs) as a function of separation between the quantum dot and source/drain leads (Wang, X., et al., Communications Physics, 3(1), 1-9 (2020). In a single quantum dot SET a measurable current only flows when the tunnel separation is ~ 12 nm or less between source/drain and the quantum dot. 20 nm separations yield $\gg G_0$ tunnel junction resistance and no measurable current. This behavior has been validated in double dot devices [<https://meetings.aps.org/Meeting/MAR21/Session/E29.2>].

In our system, we do not have individual dot control in the 3x3 array and the potential landscape across the array has significant variation across the sites as a result of disorder, electron interactions, and electrostatic effects from the leads and gates. In Figure R1 we show an example of the chemical potential across the array when adding the first electron to the first and second arrays. This is a very different picture compared to a typical double dot experiment where each dot's electrostatic potential is individually controlled by gates allowing precise alignment of the chemical potentials and opening up a transport path at low source/drain bias voltage. In our system, these pathways open up as a direct result of quantum fluctuations and many-body effects (including local and long-range Coulomb interactions and their competition with hopping processes), which determine the distribution and the extent of delocalization of electrons as they are added to the array. The formula for G in turn captures that by examining whether simultaneous charge variations on sites on the left and right edges are energetically allowed.

A key theme strengthened in the revised manuscript intended to make the quantum many body physics most evident is that by increasing the dot-to-dot separations from the first array to the third array, we explicitly elucidate the effects of tunnel coupling and electron-electron many-body interactions within the array. In the limit where the hopping/interaction ratio (t/U) approaches zero, "the simple picture of electrons tunneling through isolated dots" becomes a closer approximation, albeit in the presence of significant long-range electron-electron interactions. In the first array, the lattice sites are so close to each other that the electron wavefunctions delocalize across the array, and the individual nature of each lattice site becomes

indistinguishable. This is borne out in gate-gate maps from the first array (experiment and simulation) where the charge transition lines do not have the distinct different slopes observed in Array 3 – this is direct experimental evidence that tunneling is becoming dominant, the transition from straight charge addition lines to charge addition lines that have distinct different slopes results from site specific capacitance coupling whereas in the delocalized quantum case the capacitance coupling is to addition electrons spread across the array.

3. Alternative explanations: In the discussion of the temperature dependence, the authors write “In this system, thermal broadening of Coulomb oscillation peaks at $T=12\text{K}$ becomes large enough to eliminate smaller Coulomb blockade gaps within the upper and lower Hubbard bands but not the large Mott gap in between. Such thermally activated Hubbard band formation behavior within a 2D array is a key distinction from the thermal broadening in a zero-dimensional system where Hubbard bands do not exist”. This comes before any actual description of any experimental features. The order should be reversed so the measurements are described first. What the possible explanations? How were some ruled out? The presentation in the manuscript can be improved in many places by following this convention, eliminating duplicated explanations, and reducing the overall length.

We have reordered the descriptions that the reviewer has commented on so that the discussion comes after the description of the experimental implementation and measurements. We have rewritten in the following manner: describe the experimental implementation, the measurements, observation of the results, possible explanations, how some were ruled out.

We have revised the following sentences from:

“In this system, thermal broadening of Coulomb oscillation peaks at $T=12\text{K}$ becomes large enough to eliminate smaller Coulomb blockade gaps within the upper and lower Hubbard bands but not the large Mott gap in between. Such thermally activated Hubbard band formation behavior within a 2D array is a key distinction from the thermal broadening in a zero-dimensional system where Hubbard bands do not exist. Due to the removal of the Coulomb blockade within each of the Hubbard bands at higher temperatures, charge stability domains characterizing fixed numbers of electrons in the array become ill-defined except for at the large Mott gap. Similar transition in transport from the collective Coulomb blockade regime to the two Hubbard bands regime is also observed in the simulated Coulomb oscillations at elevated temperatures (Fig. 6b).”

To:

“We observe the removal of the Coulomb blockade within each of the Hubbard bands at higher temperatures, where charge stability domains characterizing fixed numbers of electrons in the array become ill-defined except for at the large Mott gap. We also observe a similar transition in transport in the simulated Coulomb oscillations at elevated temperatures (Fig. 6b), which can be explained by a transition from the Coulomb blockade regime to the collective Coulomb blockade regime (two Hubbard bands). In this system, thermal broadening of Coulomb oscillation peaks at $T=12\text{K}$ becomes large enough to eliminate smaller Coulomb blockade gaps within the upper and lower Hubbard bands but not the large Mott gap in between. Such thermally activated Hubbard

band formation behavior within a 2D array is a key distinction from the thermal broadening in a zero-dimensional system where Hubbard bands do not exist.”

In addition, we followed this convention and improved the presentation in the main text section Tuning the Hopping Amplitude and Long-range Interactions as follows,

“It is well known²⁷ that the nearest neighbor hopping t is exponentially dependent on the lattice-constant a while the long-range e-e Coulomb interactions are inversely proportional to a . Fig. 2 (a, d, g) show STM images after patterning from three arrays, where we intentionally increase the lattice constant $a_1 = 4.1 \pm 0.3 \text{ nm}$ in the first array, to $a_2 = 6.6 \pm 0.3 \text{ nm}$ in the second array, and to $a_3 = 10.7 \pm 0.3 \text{ nm}$ in the third array. If we interpret each artificial lattice site as an ‘impurity’ atom in silicon, the lattice constant in the first, second, and the third arrays correspond to a bulk doping density of $\sim 1.5 \times 10^{19}/\text{cm}^3$, $\sim 3.5 \times 10^{18}/\text{cm}^3$, and $\sim 8 \times 10^{17}/\text{cm}^3$, spanning from above to below the critical density of a metal-insulator transition in phosphorus-doped bulk silicon. As will be described in the following, transport measurement and charge stability analysis confirm a transition from delocalized electrons displaying metallic behavior in the first array to localized electrons in the third array.

Figs. 2b, 2e and 2h are the measured charge stability diagrams from the three arrays measured at $T \sim 10\text{mK}$ base temperature (electron temperature $\sim 300\text{mK}$). In the first array, charge addition boundaries run more or less parallel throughout the entire gate-gate plane, resembling the charge stability diagram of a single metallic island in a single-electron transistor (SET). Comparing the charge stability diagrams of the first array with the second and third arrays, a key observation is that the charge stability boundaries in the third array are dominated by straight line segments of distinct negative slopes with bias triangles at avoided crossings; whereas on the gate-gate plane in the second array, the charge addition boundaries appear crossed only at negative gate voltages and evolve into smooth curves at positive gate voltages. ... i.e., the excited addition energy state separation falls below $k_B T$ so the addition energy levels can no longer be individually distinguished at the operating temperature. (See Supplementary Materials Sections S4 for the impact of hopping, long-range interactions, and disorder on simulated addition energy spectrum and charge stability diagrams and Supplementary Materials Sections S5 for the effects from decreasing the lattice constants within the simulated array.)”

4. The authors argue that there is a resonant enhancement of transport due to hybridization between two different localized charge configurations that coupled differently to the left and right gates (config. 2 and 3 on Fig. 3a), which open additional conduction paths and reference 36 is cited. Can the authors elaborate on this and provide a qualitative description of the conduction enhancement process?

Please see Fig. R3 for a schematic illustration of the charge addition configurations and the conductance enhancement process near the resonant position between charge config. 2 and charge config. 3 in Fig. 3a in the original manuscript (now Fig. 3c in the revised manuscript). The eigenenergy spectrum and many-body eigenstate charge distributions near the same resonant position can be found in Supplementary Materials Fig. S7 of the original manuscript (Fig. S8 in the revised manuscript).

As illustrated in Fig. R3, at positive detuning and zero source-drain bias, the only charge addition configuration that aligns with the source and drain Fermi level is the one that connects the

ground charge config1 and ground charge config2. Source-drain conductance paths are mediated by a ground charge addition state mostly localized at site D32 in the bottom row.

At negative detuning and zero bias, the charge addition configuration that aligns with the source and drain Fermi level is the one that connects the ground charge config1 and ground charge config3. Source-drain conductance paths are mediated by a ground charge addition state mostly localized at sites D11 and D12 in the upper row.

At zero detuning and zero bias, the charge addition configurations that align with the source and drain Fermi level include those that connect the ground charge config1 with both the ground charge config 2 and with ground charge config 3. All three configurations become degenerate. Besides the aforementioned source-drain conductance paths that are separately mediated by the two ground charge addition states in the bottom row and the top row, additional conduction paths are allowed through resonant tunneling of the addition electron between the mostly localized charge addition configurations in the bottom row and in the upper row.

Figure R3. Supplementary figure illustrating the single electron conductance paths near the resonance (zero detuning point) between the charge config. 2 and charge config. 3 in Fig. 3a, assuming zero source-drain bias. Note Fig. 3a in the original submission has been reorganized as Fig. 3c in the revised version. Here we use the colored clouds to represent the spatial distribution of the addition electron according to the charge occupation numbers that are overlaid on the red-blue color maps in Fig. 3c of the revised manuscript. Specifically, the colored clouds at positive detuning (left panel) and negative detuning (right panel) correspond to the charge addition configurations (in Fig. 3c of the revised manuscript) moving from gate-gate position 1 to position 2 and from position 1 to position 3, respectively. The black arrows indicate viable source-drain conduction paths. At zero detuning, ground eigenstates at positive detuning and at negative detuning become degenerate and hybridize (see Fig. S8 in the revised supplementary materials). The middle panel illustrates the expected spatial distribution of the addition electron at zero detuning, which allows additional conductance via resonant transport between the upper and lower rows.

To further elaborate on the resonance enhancement in transport and provide a qualitative description of the conduction enhancement process, we added a new set of measurement data in Fig. 5 of the revised manuscript and revised the following discussion in the revised manuscript section Resonant Tunneling Through Many-body States.

“... the resonance conditions within the array enhance conductance through the array by opening additional conduction paths. Fig. 5b shows a histogram of the positive slopes of conductance resonance lines extracted from the charge stability diagrams of Array #3 over a bias range from +10mV to -10mV. An example of the bias evolution of a positively sloped resonant conductance region is shown in Fig. 5c. We attribute the difference in bias triangles at positive and negative biases to disorder in the array. From the histogram in Fig. 5b, we observe positive slope distributions centering at three distinct values representing resonant tunneling between many-body states that are primarily localized at the first and the second rows, at the first and the third rows, and at the second and the third rows of the quantum dot array, respectively. Examples are ground states at positions 2 and 3 in Fig. 3c with an avoided crossing of charge addition lines between them. The slopes of the two crossing charge addition lines (dashed lines in Fig. 5d) correspond to charge addition to sites in the upper row and to sites in the lower row, respectively. ...”

5. Can the authors explain more about why there are crossings at lower occupation numbers in Fig 4b (below the mott gap), but not on the higher occupation numbers (above the mott gap)?

Crossings are a signature of hybridization between more localized states where both the effective long-range interactions and hopping amplitudes are relatively weak. At higher occupancies, there is increased screening with additional electrons in the array. The energy landscape within the array becomes smoother and the added electron spreads out more widely on the array. The absence of crossings at higher occupation numbers above the Mott gap is a signature of increased quantum fluctuations and that the many body states are becoming more delocalized over the array.

6. Do you see transport closer to zero bias, or just at several mV biases (Fig. 2a, 2b, and Fig. 4b, 4e)? Those are very high biases.

Yes, we see transport closer to zero bias in all three arrays. This is evidenced by the measured Coulomb diamond diagrams in Fig. 2c, Fig. 4c, 4f (original manuscript) and Figs. 2f, 2g, and 2h (revised manuscript) where finite conductance is visible at close to zero source-drain bias V_{DS} . The example below, Figure R4, shows a conductance map of Array #3 at +0.1 mV bias, where transport is visible but discontinuous at some locations.

Figure R4. Conductance gate-gate map of Array #3 at +0.1mV bias.

7. Why are the calculations done at $T = 1 \text{ K}$, given that the data was taken at base temperature?

We modified the caption of Fig. 2e in the original version (Fig. 3b in the revised version), where the numerically simulated conductance gate-gate map is calculated at $kT = 1 \text{ meV}$ rather than $T = 1 \text{ K}$. The reasoning behind our choice is as follows.

Effective electron temperatures are at least 300 mK in these experiments, which is the relevant temperature for the calculations. Base temperature, while relevant, is only the bath temperature. The experimental conductance maps were measured at finite bias voltage of the order of a few mV and the extent to which excited many-body states participate in transport is largely determined by the bias window, rather than thermal broadening. The calculated conductance does not take into account the effect of the environment, including the leads and the bias window, or other processes, such as phonons, which can further broaden the measured conductance peaks even at very low temperatures. Therefore, we approximate these effects and the participation of excited states in transport by setting a 1 meV thermal broadening in the numerical calculations. In Fig. 6b, we show the conductance for array #2 along the diagonal direction in the gate-gate map at temperatures as low as 0.6K and on a finer grid for gate voltages. The narrow peaks seen there are expected to become even narrower at lower temperature.

We have added relevant information into the method section of the revised main text.

“The experimental conductance gate-gate maps were measured at finite bias voltage on the order of few mV and the extent to which excited many-body states participate in transport is largely determined by the bias window, rather than thermal broadening. The calculated conductance does not take into account the effect of the environment, including the leads and the bias window, or other processes, such as phonons, which can further broaden the measured

conductance peaks even at very low temperatures. Therefore, we approximate these effects and the participation of excited states in transport by setting a 1 meV thermal broadening in the numerical calculations of conductance gate-gate maps of Fig. 3b in the main text and Fig. S5b in the supplementary materials.”

8. There is a discussion about non-closing diamonds in Fig. 2c (line 263). It would be very helpful for the reader if the authors point out exactly which diamonds they are referring to, and where they predict the closings would be. As is, it is a bit vague, because we don't know exactly which diamonds the authors refer to. That also makes it hard to understand lines 273-277. Additional description would be helpful.

We have revised Fig. 2c in the original version (Fig. 2i in the revised version) with circles that mark examples of open and closed Coulomb diamonds that are described in the main text.

9. “The middle row” -> It is explained that the charge configurations involving the middle row, with an intermediate slope, can be seen in the calculations in Fig. 2d. Are these observed in the data? If not, why are they not observed in the data?

In Figure R5, the left panel is a reproduction of Fig. 2d from the original manuscript, and the right panel plots the charge transition boundaries with the small, intermediate, and large slopes plotted in orange, green, and blue respectively. For observations of the intermediate slopes in the experimental data, please see the slope analysis for the measured conductance lines in Fig. 6 (a, d,g) in the original main text (Fig. 4(c, f, i) in the revised main text). The presence of intermediate slopes is evidenced by the middle peak in the histogram distributions of slope based on analysis on both the experiment and theory data.

Figure R5. Left panel: reproduction of Fig. 2d in the main text. Right panel: extracted charge addition boundaries with the small, intermediate, and large slopes plotted in orange, green, and blue respectively.

10. Consider re-arranging to present the data in the following order: metallic limit, then intermediate, then multi-island last, because the multi-island measurement has additional description that is distracting. The main message would come through first, in this different order.

We have made significant rearrangements in the manuscript to first present the metallic limit, then intermediate and multi-island last. This undertaking resulted in major re-organization of the manuscript. However, we do believe this has significantly improved the presentation and flow. Please note that we introduce the three arrays in this new order and have therefore also moved the tunability of hopping amplitudes and long-range interactions by changing the lattice constants within the array to before demonstrating that we can define the electron ensemble, characterize the charge addition boundaries and resonant tunneling within the 2D array, and tuning of the hopping.

11. In my opinion, some of the figures can be put in the supplementary info. For example, Fig 5 can go there and Fig 6 can go there. It is enough to summarize the results in main text.

We have moved Fig. 5 into the supp materials in the rearranged revision. We believe that Fig 6 remains an important figure to both elucidate the capacitive control of dots within the array and build confidence in the understanding of the array physics as Fig. 6 shows good qualitative agreement between the model and experiment. Fig. 6 in the original manuscript is now Fig. 4 in the revised manuscript.

12. It is claimed in lines 273-277 that the asymmetry of Coulomb diamonds and non closing of some of them in Fig2C, is mainly due to the variation in binding energy. To support this claim, is it possible to extract a rough estimate of the binding energy variation from the data in Fig2C?

Figure S8 and the associated paragraphs in the supplementary materials in the original manuscript (Figure S9 in the revised manuscript) described the determination of binding energy levels from the measured Coulomb diamond diagram in Fig.2c of the original manuscript (Fig. 2i in the revised manuscript). Here we treat the array as a many-body system and plot the extracted binding energy levels in Fig. S8c of the original manuscript (Fig. S9c in the revised manuscript) for both the positive and negative bias conditions. As illustrated in detail in our response to Reviewer #1, single electrons conduct through many-body states of the array rather than isolated, individual quantum dots. Therefore, the measured binding energy spectrum in Fig. S8 does not represent binding energy distributions of individual array sites.

When treating individual array sites as isolated quantum dots, we refer the reviewer to the quoted values in Table S3 Column 2 as a rough estimate of the binding energy variations of the individual array sites. The variation of on-site binding energies, in combination with many-body interactions within the array, results in an inhomogeneous chemical potential landscape within the array, explaining the asymmetry of Coulomb diamonds and non-closing diamonds in Fig2c of the original manuscript (Fig. 2i in the revised manuscript), as being mainly due to the variation in binding energies.

13. It is stated that inside their quantum simulator that applying a common voltage on both in-plane gates can be used to sweep the array's global chemical potential (line 177-178). However, this is unlikely to be the case, because the source and drain will screen the potential and make it non-uniform. Please include a more nuanced discussion of this.

We have modified the description on page 6 line 177 to be more complete and accurate.

“The number of electrons in the array ($\sum_i n_i$), which determines the filling factor, can be altered by applying a common voltage on both in-plane gates to sweep the array’s global chemical potential with respect to the Fermi level.”

New wording:

“The number of electrons in the array ($\sum_i n_i$), which determines the filling factor, can be altered by applying a common voltage on both in-plane gates to shift the array’s chemical potential landscape (albeit non-uniformly due to screening from source and drain leads) with respect to the Fermi level.”

Minor comments/questions

1. “readily achievable zero-temperature limit” on line 53. Something is wrong with this statement, since zero temperature cannot be achieved.

We have corrected the sentence to be more accurate,

“readily achievable low-temperature-limit with respect to the hopping amplitude.”

2. One advantage of the STM fabrication technique used by the authors is that it can place dopant atoms with ~ 1 nm precision. This is true. It is stated that ion implantation techniques are limited in resolution to 30-50 nm, online 63. However, higher implantation accuracies are possible, there is some literature suggestion ~ 5 nm is possible. Please include a short but more nuanced discussion of this.

We have modified the original sentences to a more nuanced discussion regarding the limitations of ion-implantation technique in atomic precision dopant placement and referenced the latest implant research indicating 5 – 10 nm placement.

Previous wording:

“The Anderson-Mott transition has been previously demonstrated in few-atom systems using ion-implanted single dopant impurity atoms in silicon. However, the dopant positioning accuracy using the ion-implantation technique has been limited to $\sim 30 - 50$ nm.³ Effective control of tunable Hamiltonian parameters and precision-engineering of electron and spin correlations in a dopant-based artificial lattice relies on the controlled placement of dopant atoms in the host lattice with near-atomic precision. ”

New wording:

“Effective control of tunable Hamiltonian parameters and precision-engineering of electron and spin correlations in a dopant-based artificial lattice relies on the controlled placement of dopant atoms in the host lattice with near-atomic precision. Although the Anderson-Mott transition has been previously demonstrated in few-atom systems using ion-implanted single dopant impurity atoms in silicon,³ the ion-implantation technique is limited by implantation aperture size and

ion straggle. To our knowledge the best reported implant positioning accuracy is $\sim 5 - 10 \text{ nm}^4$ which is incompatible with multiple site atomic-precision dopant arrays.

3. It is stated that the STM-based hydrogen lithography was proposed by some authors and extensively developed at UNSW, on line 68. I think it is more accurate to say that it was developed, and expanded into a technique enabling the placement of dopant atoms on a surface at UNSW, and covering those with a thin layer of silicon

To describe the technical development history more accurately, we have revised the original sentences in line 68,

Original text:

“Atomic-scale precision dopant placement can be achieved using the STM-based hydrogen lithography technique in silicon, initially pioneered by Joseph Lyding’s group⁵ and extensively developed by Michelle Simmons’ group at UNSW,^{6,7} and more recently by groups at NIST,^{1,8} UCL,^{9,10} and Sandia.^{11,12}”

The new text reads:

“The STM-based hydrogen lithography technique, initially pioneered by Joseph Lyding’s group,⁵ was further developed and expanded by Michelle Simmons’ group at UNSW^{6,7} into a method that allows controlled dopant atom placement buried in an epitaxial Si environment enabling a new suite atomic-scale precision devices. Several groups have further developed this atomic-scale device fabrication technique introducing new applications, e.g. NIST (dopant-based analog quantum simulation),^{1,8} Sandia (CMOS integration),^{11,12} and UCL,^{9,10}.”

4. The authors use the convention that a positively charged site is called D+, a neutral site is called D0, and a negative charged site is called D- (for example, line 145-148). This is the convention for single donors in the literature. I am not aware that this is used to describe multi-donor quantum dots. I don’t think this convention should be used here.

Following the reviewer’s comment, we have removed the D0, D- and D+ notation in the main text line 147.

“... charge number fluctuations of up to two electrons at each site which corresponds to the three charge states of a few-dopant quantum dot: the ionized state, the charge-neutral state, and the negatively charged state.”

In line 193

“A quantum dot size of 2 ± 1 dopants corresponds to an on-site electron-electron interaction energy U_i of $\sim 45 \text{ meV}$ (for the relevant charge-neutral to negatively charged state transitions).”

And in line 718

“Table S3. Binding energy E_b and addition energy U_i (for the relevant charge-neutral to negatively charged state transitions near the Fermi level) of few-P cluster quantum dots.”

5. Missing citation to go with “We ignored Coulomb exchange and higher-order hopping terms in our numerical model” on line 170.

Calculations of Le et al. have verified that long range tunneling terms and additional interaction terms other than the on-site and inter-site Coulomb repulsion terms are insignificant for calculations of the ground state properties of an ordered dopant array, as well as the order of magnitude of the conductance of a disordered dopant array. We have added the missing citation (Le, et al. PRB 2017) in the revised version.

6. To my knowledge, Anderson localization (line 215) involves the localization of non-interacting electrons, while the Anderson/Mott localization describes the transition for interacting electrons.

We agree with the reviewer’s comment. We have corrected the terminology.

“Variations in the number of dopant atoms and detailed dopant cluster configurations at each lattice site introduce site disorder, which in larger arrays lends itself to the study of Anderson/Mott localization, and especially its existence in the presence of strong interactions.”

7. Line 220-221: “changing the lattice constants within the array” -> “measuring arrays fabricated to have different lattice constants”.

We have modified the sentence according to the reviewer’s suggestion.

“Then we show tuning of the hopping amplitudes and long-range interactions by measuring arrays fabricated to have different lattice constants.”

8. Line 235: “at a base temperature of $T \sim 10$ mK” -> What is the electron temperature?

We characterize the electron temperature by measuring the thermally/noise broadened Coulomb oscillation peaks in a single electron transistor. At the base temperature of the fridge, the estimated electron temperature is ~ 300 mK. We have added the electron temperature in the main text.

“Figs. 2b, 2e and 2h are the measured charge stability diagrams from the three arrays measured at $T \sim 10$ mK base temperature (electron temperature ~ 300 mK). ... ”

9. The parallel and serial transport configurations discussed in lines 269-270 are not well explained at that point. Are these the transport paths shown in Fig. 3 and Fig. 4?

Our original intention of mentioning the coexistence of the parallel and serial transport configurations in the sentence mentioned was to highlight the complexity in possible

conductance paths through the 2D array. After careful reconsideration, we agree with the reviewer that the original statement could be misleading in the sense that “parallel and serial” transport configurations only apply to arrays in the atomic limit. In the metallic limit where many-body state wavefunctions spread across the entire array, transport through the array can not be described by the language of sequential tunneling. We have removed the sentence in question from the revised main text.

10. The sentence “indicating a quasi-continuous (metallic) density of state distribution in the third array” on line 407 deserves further clarification. Does this just mean that the excited state separation falls below $k_B T$ so the states can no longer be individually distinguished at the operating temperature?

Yes, the reviewer’s interpretation is correct. We added clarification in the main text as follows.

“In the second array (Fig. 2f), resonant conductance is visible as lines of increased differential conductance running parallel to the edges of Coulomb diamonds, indicating a discrete eigen-energy spectrum within the array; such a discrete conductance spectrum is not visible in the first array (Fig. 2c), indicating a quasi-continuous (metallic) density of state distribution in the first array, i.e., the excited addition energy state separation falls below $k_B T$ so the addition energy levels can no longer be individually distinguished at the operating temperature.”

11. The sentence on line 443 should be clarified, why is the word melting used? Melting seems to imply something thermal. But nothing thermal happens when two parameters in a Hamiltonian vary with respect to each other. Some energy scale needs to be compared to temperature.

The reference to “melting” has been removed from the manuscript to avoid confusion. Indeed, we are not referring to a thermal process.

12. The comment about the Wigner-like phase on line 457 is counterintuitive to me. Wigner localization refers to spontaneous localization of electrons due to electron-electron Coulomb repulsion. But in a system with attractive ions, there is an explicit orbital localization due to the electron-ion attractive. Please explain what this has to do with Wigner localization.

The referee is correct that strictly speaking, a Wigner crystal refers to spontaneous arrangement of electrons due to interactions not necessarily in the presence of any underlying lattice or attractive centers. Hence, the word “Wigner-like”, which in Ref. 32 [Prati, E., Hori, M., Guagliardo, F., Ferrari, G. & Shinada, T. Anderson–Mott transition in arrays of a few dopant atoms in a silicon transistor. *Nature Nanotechnology* 7, 443–447 (2012)] has been used to describe electrons being frozen at dopant sites because of long-range Coulomb repulsions as well as local attractions.

13. Please explain the source of asymmetry in the line shape for the data in Fig3C.

In Fig. 3c (now Fig. 5e), we fit the series resonant tunneling through two discrete many-body ground states that maximally hybridizes at zero detuning. The increased background conductance at negative detuning direction arises from additional conductance channels at

large source-drain bias windows, which could include higher order resonant tunneling between ground and excited discrete many-body states, inelastic processes, and co-tunneling.

14. The authors present the calculated charge configurations in Fig. 3a and 5c. Do the authors have any experimental data (perhaps from nearby charge sensors, if any) to support the calculations?

In this study, we extract information of charge configuration within an array from experimentally measured charge stability diagrams that are mapped by monitoring the direct current transport through the array while sweeping in-plane gate voltages. We do not have nearby charge sensors built into the array devices in this study, although we are currently developing remote charge sensing at NIST for future studies. We take the best experimental estimates of disorder and hopping amplitude as inputs to generate the calculated charge stability diagrams shown in Fig. 2(d, e) and Fig. 5(a, b) in the original manuscript (Fig. 3(a, b) and Fig. S10(a, b) in the revised manuscript), which are supported by their qualitative agreement with the experimental charge stability diagrams in Fig. 2(a, b) and Fig. 4(b, e) in the original manuscript (Fig. 2(h) and Fig. 2(e, b) in the revised manuscript). Since the exact disorder configuration is unknown, it is not our intention to exactly map the calculated charge configurations directly to the experimental results. Instead, we extract representative charge configurations Fig. 3c and S10c in the revised manuscript to qualitatively illustrate the trend in many-body state hybridization and delocalization when tuning the lattice constants.

Reviewer #3 (Remarks to the Author):

The paper describes the electrical transport study of a small array of single/few dopant atom quantum dots in silicon, where the number of dopants per dot was nominally 1-3 but varied randomly, with limits in dopant number set by the size of the 'hole' in the hydrogen resist mask used to fabricate the array. Numerical simulations of the data provide insight into the mechanisms responsible for the transport phenomena observed. The work is of interest to a broad readership, and is extremely important, as it demonstrates a clear advance towards studying many body physics using atomic-scale dopant-in-silicon structures, made using the STM based hydrogen lithography method. This field has been evolving for the last 20 years, and recently resulted in the demonstration of a 2-qubit qubit gate, but still has not revealed much new fundamental physics. The study here paves the way for that to change, demonstrating the quantum simulation of an extended 2D Fermi-Hubbard Hamiltonian. The work is impressive and thorough, although some of the discrepancies between experiment and simulation (that I would expect) are brushed over a little too easily, as noted below. If the relatively minor issues highlighted below are resolved, then I highly recommend that the paper is published.

In general: While there is generally reasonable qualitative agreement between theory and simulation, and the simulation is invaluable for explaining the fundamental physics at play, the agreement between experiment and simulation is far from perfect. For example, Fig. 2a and 2b have the same trends as Fig. 2e, but there are also significant differences. Likewise, Fig. 6d-i. This seems perfectly reasonable to me, as the actual numbers of dopant in each dot in each array is an educated guess rather than a known value. However, this is not sufficiently highlighted in the paper. The obvious place to do this would be in the discussion section. I would like to see a paragraph discussing how you would expect the agreement between theory and experiment to improve (which I presume it would) as the number and position of the dopants in the array became more accurate controlled, with the ideal situation of 1 dopant in each

dot at a perfectly precise location. How much more closely would you expect the data and theory to agree in that situation.

We added the following paragraph to the discussion section in the main text, between the first and second paragraphs in the discussion section.

“In this first demonstration of analog quantum simulation using atomically patterned 2D arrays of dopants, the quantum simulation accuracy is limited by uncertainties in the exact number of dopant atoms and their clustering configurations within each array site, as well as atomic-scale variations in nearest-neighbor hopping amplitudes. Continuing efforts are underway within our group and in the research community to seek long term solutions in improving the atomic perfection in dopant-based quantum devices. In a separate study, [Wyrick, J. et al. Enhanced Atomic Precision Fabrication by Adsorption of Phosphine into Engineered Dangling Bonds on H-Si Using Scanning Tunneling Microscopy and Density Functional Theory. arXiv preprint arXiv:2112.12200 (2021)] we demonstrated fabrication of a 6-site array with precisely one single dopant per site by combining low-temperature feedback-controlled hydrogen lithography with a new method called feedback-controlled manipulation. The precise number of dopant atoms in a few-dopant quantum dot and its clustering configuration can be measured using STM spectroscopy after dopant incorporation and before Si epitaxial overgrowth. Combining with ab-initio calculations of the electronic structure of measured clusters as inputs, we expect significant improvement in the agreement between the experiment and theory in the future. In addition, development of more complex virtual gate designs is underway towards addressing individual chemical potential at each array site and fine tuning of local tunnel coupling, enabling full control of the dopant-based analog quantum simulators.”

In general: The paper discusses in great detail the uncertainty in the number of dopants in a dot. However, was the position of a dopant within a dot considered? For example, if D12 in array 1 had two dopants in it, the dopants could be ~ 0.4 nm apart or ~ 2 nm apart. How might this difference affect the transport data, if at all?

The reviewer has correctly pointed out an important source of uncertainty within the arrays in addition to the number of dopants per dot, namely the uncertainty in the exact dopant configurations within each few-dopant quantum dot. The relevant literature for calculations, [Weber, B. et al. Spin blockade and exchange in Coulomb-confined silicon double quantum dots. Nature Nanotechnology 9, 430–435 (2014)] for a given number of dopants in a dot, with variable clustering configurations introduces variation in binding energies for the dot, and therefore the chemical potential landscape within the array. The one sigma values of the expected binding and charging energy values that we included in Table S3 in the supplementary materials characterize energy scale variations due to probabilistic distributions of cluster configurations. Another effect of different cluster configuration is the change of electronic structure and molecular orbital wavefunctions at an array site, and the site-to-site tunnel coupling. Under the experimental conditions in this study, we expect the effect of clustering variations on transport data to be similar to, and experimentally indistinguishable from, the transport effect due to uncertainties in the number of dopants. To clarify this argument, we have added the following sentences to the paragraph on page 7 line 208.

“Instead, we account for the effects of disorder by estimating the number of dopant atoms-per-site, based on STM-lithography patterns and dopant incorporation conditions, for use as input to the numerical simulations. We do not distinguish the effects from different sources of disorder because variation in dopant number per site and their clustering configuration have similar effects on site-by-site energy variations. While we have not pursued the exact match between the numerically simulated and experimentally simulated results in this study, the detailed atomic configuration at each array site does not alter the qualitative understanding of the array system (see Supplementary Materials Section S6), and the quantitative differences between theory and experiment are a measure of the accuracy of the disorder estimates.”

Specific comments to be addressed:

Page 8, Fig. 2e: Why has a different voltage range been chosen compared with the experimental data? I suspect it is so theory and experiment look more similar, but it does not make sense to do this. Please correct this, or state in the text why this is not done. Also, the red ellipses added to Fig. 2b and 2e are almost invisible when printed. These should be made clearer.

Please see the updated Fig. 2 and Fig. 3 in the revised main text. All simulations and experimental data are now shown for the same gate ranges. The red ellipses in the new figures have been made clearer.

Page 10, after line 304: For clarity, the authors should explain in a bit more detail the physical interpretation of Fig. 3a. For example, later in the paper it is mentioned that the charge distributions and single charge addition in Fig. 5c demonstrate greater delocalization of electrons throughout the array. However, delocalization (or lack of it), and how it is inferred from the figure, was not described when discussing Fig. 3a. It would be helpful for the reader if it was. The addition, it is hard to immediately appreciate what is happening during the addition of the 8th electron Charge Config. 1 to 2, and 9th electron Charge Config. 2 to 4. For example in the eigenstate charge distributions of Charge Config. 2, it's hard to tell if the charge occupation in D11 is different to D12, and if D12 and D23 are different. I assume so, as following the addition of the 9th electron D11 and D12 look identical to the surrounding arrays. It also looks like 1.5 electrons are being added for the 9th electron. In addition to the figure, could you describe the charge occupation in D11 and D12 (for example) in terms of percentages of an electron charge, or whatever is most appropriate?

To compare the amount of delocalization in original Fig. 3a (now Fig. 3c) and original Fig. 5c (now Fig. S10c), we added occupation numbers in the eigenstate charge distribution and single charge addition plots. By lack of delocalization, we refer to the scenario where eigenstates and charge addition occupations are (or close to) integers at each array site. By delocalization, we refer to the scenario of significant fractional-charge-occupations across multiple sites.

The added numerical values in figures also help clarify the changes in occupation numbers at each array site that may not be easily discernible due to subtle changes in color scales. As an example, for Fig. 3a (new Fig. 3c), adding the 8th electron changes the ground eigenstate charge

occupations from config. 1 to config. 2. At Config. 1, finite hybridization exists between D11 and D12 as evidenced by the fractional occupation at these two sites. When adding the 8th electron, the added electron primarily localizes at D32. Meanwhile, a small fractional change of electron occupation (0.086e-) occurs between D12 and D11, energetically favorable for the ground many-body states of 9 electrons. This leads to the charge occupations shown in the ground eigenstate charge configuration at point 2.

Please see Fig. 3 and Fig. S10 in the revised manuscript for the updated figures. Note that we have introduced substantial reorganization in these figures in order to reflect comments from all reviewers.

Page 15, line 437 (discussion of Fig. 6): The statement “Unstructured distributions at higher slopes ($\Delta V_{G2}/\Delta V_{G1} < -3$) are likely due to combined effects of defects and edge detection artifacts” is unconvincing. What defects are expected? No extra dopants should be incorporated at the individual dangling bond sites. I do not see any evidence of buried dopants in the STM image. If edge detection artifacts cause that much of an effect, can they not be reduced by more sophisticated data processing methods? Is it not a more reasonable argument to say that the distribution of the number of dopants in the array is significantly broader than you think? For example, wouldn't dots with zero dopants skew the transport data more significantly than the dots that have non-zero numbers of dopants. The comparison between experiment and theory for array 1 (Fig. 6d and 6g) is reasonably convincing, despite the noise and the narrowing of the distribution for the arrays 2 and 3 is also very crudely in agreement between experiment and theory. However, I would not describe the agreement for the latter case to be good, and this should be commented on. Again, there is just no way of knowing exactly how many dopants are in each dot in the experiment so I would not expect good agreement. The authors should either state this, or a plausible alternative.

We thank the reviewer for proposing additional possibilities for the analysis of the capacitive coupling and associated slopes. It is possible that the predicted skewing of lever arm ratio distributions can be modified in the proposed scenario where the distribution of number of dopants per site is broader than estimated, including the possibility of zero dopants at certain sites. However, we expect that this is partially due to variation in gate control for each individual dot – all 3 dots in a row are not exactly, equally coupled to the leads due to screening from other dots and electrodes. Also, the leads and their geometries are not perfectly symmetric at the atomic scale - each of these will broaden the distribution of slopes. Further, gate non-linearity will cause broadening of the slopes as well. We have revised our original description in the context of the actual data acquisition conditions. The following revised wording has been added to the text on page 15 line 437. We also added comments regarding the agreement between the theory and experiment.

“The extended skewing in slope distribution towards more negative slopes ($\Delta V_{G2}/\Delta V_{G1} < -3$) is likely the result of imperfect symmetry in device geometry and gate nonlinearity. Similar results are found in Figs. 4g, 4h, and 4i plotting the distributions of the lever arm ratios of charge additions, which are calculated from the simulated charge stability diagrams using the identical

bias windows as in the experimental plots to include charge addition configurations of excited many-body states. (See Supplementary Materials Section S1 for detailed capacitance and lever arm calculation methods). We note that the agreement in the shapes of the slope distributions between experiment and theory remains qualitative due to unknown disorder at the atomic scale and real measurement conditions that are not fully accounted for in theory.”

Page 25, Fig. S2d: For D13 in array 3, why is the lower bound assigned as 1 dopant (Table S2)? This ‘dot’ appears to be made from 2 regions of isolated desorption, with each region consisting of an identical line of 4 dangling bond (DB) pairs. Either you assume that 4 DB pairs have the possibility of resulting in zero dopants, in which case the lower bound is zero, else the assumption is that at least 1 dopant must be incorporated at 4 DB pairs, in which case there will be 2 dopants in total.

We thank the reviewer for catching this point for D13 in array 3. We agree with the reviewer that, based on our criteria of estimating the lower bound of dopant numbers, the lower bound should be 0 for each desorption region, and the lower bound should still be zero in total for D13. We have made the correction in Table S2 of the revised manuscript. At the same time, the best estimate and upper bound of the dopant numbers for D13 are not affected.

Minor amendments:

Page 4, Fig. 1a: As this schematic is earlier in the paper than discussions of adding electrons to the array, the reader might try to correlate the number of valence electrons (shown as spins) on the quantum dots with the number of dopants in the dots. However I assume that the number of electrons in the array has been arbitrarily assigned for the schematic. This should be clarified in the caption. I would have liked to have seen the electrodes labelled also (Source, Drain, Gate), but I leave this to the discretion of the authors as it might then look less visually appealing.

We added an additional sentence to Fig. 1a caption. We also added labels to the source, drain, and gate electrodes in the schematic.

“Figure 1. 2D dopant-based quantum dot arrays as a platform for simulating the extended Fermi Hubbard model. (a) Schematic of the experimental Fermi-Hubbard system composed of a 3x3 array of single/few-dopant quantum dots coupled to in-plane gates and source-drain leads, allowing transport measurements through the array. The number of electrons (shown as arrows) and dopant atoms at each array site (pink dots) in this schematic have been arbitrarily assigned for illustration purposes.”

Page 8, line 235: You should state that it is array 1 that you are discussing. It is not entirely obvious from the ending of the previous section (although not a great surprise!).

In the new organization of the manuscript, we added a first sentence to the section “Gate-tuning the Electron Ensemble and Charge Distributions” to clarify that the discussion in this section is based on results from the (now) third array.

“In this section, we use results from the third array, which is near the atomic-limit (weak tunnel coupling limit - $U/t \gg 1$) to illustrate tuning of the charge distributions within an array using in-plane gates. At a base temperature of $T \sim 10$ mK (electron temperature ~ 300 mK), we measure the transport spectrum through...”

Page 12, Fig.5, caption: For (c) it should be mentioned that it is array 2 that is being discussed.

Please note Fig. 5 in the original manuscript was moved to the supplemental material based on a reviewer suggestion. This figure is now Fig. S10 in the revised manuscript. The change to the figure's caption is highlighted in red.

“(c) Schematic illustration of simulated eigenstate charge distributions and single charge addition for the second array charge stability diagram region as highlighted by the red box in the left diagram in (a). Note the significant increase in electron delocalization as the lattice constants are reduced. The ground state charge distribution and charge addition plots following the same convention as described in Fig. 3a.”

Page 19, Line 582: The reference should be Ref. 27 not Ref. 24 and it is Le et al.,

We have corrected this reference number in the final revision.

Figure 25, Figs. S2c and S2d: Enlarge the arrays (which are the important part for the associated discussion) in the middle of the image, cropping the images significantly if necessary.

We have enlarged the central parts of Figs. S2c and S2d with additional cropping to more clearly present the important parts of the STM images for the second and the third arrays. We have also updated the figure caption as shown below.

Figure S2. Estimating the number of incorporated dopant atoms at each quantum dot based on atomic-resolution STM images of hydrogen lithography patterns before phosphine dosing. (a) (b) (c) Atomic resolution STM image of the central region of the first array device (a), the second array device (b), and the third array device (c) after hydrogen lithography but before phosphine dosing, overlaid with surface lattice grids and identifiers of desorbed dangling bonds and dimers at each array site. (d) Zoom-in images at each quantum dot in (c) with overlaid grids of Si(100) 2x1 surface reconstruction unit cells and identifiers of desorbed dangling bonds and dimers at each array site.

Page 26: The font size changes on line 708.

In the final revision, we will ensure the font size remains consistent throughout the manuscript.

Figure 27, Figs. S3c and S3d: It is hard to read the text or see the arrows clearly when the paper is printed. This should be rectified.

We have rectified Figs. S3c and S3d to make the text and arrows easier to read.

REVIEWERS' COMMENTS

Reviewer #1 (Remarks to the Author):

I agree with the argument that the conductance would be much smaller if the electrons tunnelled through only individual dots. The relatively large conductance observed is thus a strong evidence of tunnelling through many-body states. While an experiment with the capability for controlling individual dots would provide a clearer picture, this work will attract a lot of interest and is suited for publication in Nature Communications.

In a few places in the main text the authors used "last author and coworkers" or "last author et al" to refer to other works, for example on pages 18 and 19. The convention "first author et al" should be used to avoid confusion.

Reviewer #2 (Remarks to the Author):

All of the referees agreed that the experiment represents a significant advance in the field of dopant-defined quantum dot arrays. The questions from the three referees mainly concern three separate points summarized below.

- 1) Is this a quantum simulator?
- 2) Can this be a quantum simulator in the future?
- 3) Is the system 2D and is there physical evidence for many-body quantum states?
- 4) Other questions about the experiment

The authors have made extensive revisions to the manuscript to address the questions from the referees.

In summary, upon reading the reply of the referees and reading the revised manuscript, my view is the following

1) The current experiment is not a quantum simulator. There is a very high degree of uncertainty in the underlying Hamiltonian. As highlighted by referee #1, this is a Coulomb blockade experiment

2) It is possible that if a number of hurdles are overcome, dopant atoms can eventually become quantum simulators of 2D states. These challenges include (a) atomic scale knowledge of the position of the dopants, (b) ability to sufficiently manipulate or measure the system (more difficult for dopants than quantum dots), (c) taming the valley physics or getting rid of valleys by using acceptor atoms.

3) The main argument used to establish that the system is 2D is that quantum dots in the “middle” row (row farthest from both the source and drain) row are too far from source and drain leads to be involved in direct transport. I do agree that it means transport is being mediated by all atoms in the array. The argument that that different slopes in single-electron transport represent columns of dopant atoms acting as molecules is certainly plausible. If it is true, this would imply resonances at their ant-crossings involve superpositions of different columns.

4) The revisions made to the manuscript are helpful in clarifying key technical aspects that needed to be addressed for the article to be suitable for publication.

My conclusion is that this is solid work. However, the authors cannot claim they have done a quantum simulation. There is definitely evidence that tunnel coupling impacts the Coulomb blockade transport in this system. If the authors modify the title, abstract, and claims of the paper to avoid over-claiming 2D quantum simulation, and focus on claims that can be justified in Coulomb blockade transport, then the article would most likely be suitable for publication.

Reviewer #3 (Remarks to the Author):

The authors have successfully changed the manuscript such that my comments and concerns have been addressed. I am happy for publication to proceed.

REVIEWER COMMENTS

We thank the reviewers for their comments. We have very carefully considered the reviewers remaining concerns and comments and have addressed these in the current submission. We have also provided a detailed point-by-point response below to the reviewer's comments.

Reviewer #1 (Remarks to the Author):

I agree with the argument that the conductance would be much smaller if the electrons tunnelled through only individual dots. The relatively large conductance observed is thus a strong evidence of tunnelling through many-body states." While an experiment with the capability for controlling individual dots would provide a clearer picture, this work will attract a lot of interest and is suited for publication in Nature Communications.

We thank the reviewer for the comments and agree with your interpretation. We are in the process of making new fully gated arrays and arrays with near atomic perfection using the methods described recently in Ref. 43.

In a few places in the main text the authors used "last author and coworkers" or "last author et al" to refer to other works, for example on pages 18 and 19. The convention "first author et al" should be used to avoid confusion.

We now use the convention "first author et al" throughout.

Reviewer #2 (Remarks to the Author):

All of the referees agreed that the experiment represents a significant advance in the field of dopant-defined quantum dot arrays. The questions from the three referees mainly concern three separate points summarized below.

- 1) Is this a quantum simulator?
- 2) Can this be a quantum simulator in the future?
- 3) Is the system 2D and is there physical evidence for many-body quantum states?
- 4) Other questions about the experiment

The authors have made extensive revisions to the manuscript to address the questions from the referees.

In summary, upon reading the reply of the referees and reading the revised manuscript, my view is the following

1) The current experiment is not a quantum simulator. There is a very high degree of uncertainty in the underlying Hamiltonian. As highlighted by referee #1, this is a Coulomb blockade experiment

We agree with the reviewer that the disorder in our array system introduces significant uncertainty in the underlying Hamiltonian. We have added the following wording to the shortened abstract “we fabricate 3×3 arrays of single/few-dopant quantum dots with finite disorder” and to the Introduction “although disorder present in the arrays leads to uncertainty in the underlying Hamiltonian”. Disorder is also mentioned several times in the manuscript including the previous existing full paragraph in the Discussion Section on the effects of disorder which says “accuracy is limited by uncertainties in the exact number of dopant atoms and their clustering configurations within each array site”.

In response to Reviewer 2 comments, and in the interest of a more conservative statement of the claims, we have changed the title of the manuscript and removed all explicit claims to having demonstrated an analog quantum simulation. We do, however, believe that the following supports that we have demonstrated both the experimental realization of an extended Hubbard model Hamiltonian and that we have tuned key parameters in the Hubbard Hamiltonian:

1) By tuning of the electron ensemble using in-plane gates and low temperature transport, we have characterized the charge stability configurations, addition energy spectrum, and the impact of disorder.

2) By controlling the array lattice constants with sub-nm precision, we demonstrated tuning of the hopping amplitude and long-range interactions and observe the finite-size analogue of a transition from metallic to Mott insulating behavior.

3) By increasing the measurement temperature, we simulated the effect of thermally activated hopping and Hubbard band formation in transport spectroscopy.

2) It is possible that if a number of hurdles are overcome, dopant atoms can eventually become quantum simulators of 2D states. These challenges include (a) atomic scale knowledge of the position of the dopants, (b) ability to sufficiently manipulate or measure the system (more difficult for dopants than quantum dots), (c) taming the valley physics or getting rid of valleys by using acceptor atoms.

Although we have removed all explicit claims of having demonstrated an analog quantum simulator, we have made progress towards a more accurate representation of the Hubbard Hamiltonian. As mentioned above in the response to Reviewer 1, we have a manuscript with preliminary acceptance to a leading journal which demonstrates single atom precision using STM fabrication (Ref. 43). We are in the process of using this new method to fabricate arrays with precisely 1 dopant atom per site.

We have also recently demonstrated and presented at the 2022 Silicon Quantum Electronics Workshop, RF reflectometry measurements of a 2x2 array showing single electron position sensitivity in the array, adding an additional measurement capability. We are currently investigating valley physics in our array system and its effects on spin filling measurements, however, this is beyond the scope of the current manuscript.

3) The main argument used to establish that the system is 2D is that quantum dots in the “middle” row (row farthest from both the source and drain) row are too far from source and drain leads to be involved

in direct transport. I do agree that it means transport is being mediated by all atoms in the array. The argument that that different slopes in single-electron transport represent columns of dopant atoms acting as molecules is certainly plausible. If it is true, this would imply resonances at their ant-crossings involve superpositions of different columns.

We agree with the reviewer's interpretation and believe that the revised manuscript makes a compelling case for transport through the many body states.

4) The revisions made to the manuscript are helpful in clarifying key technical aspects that needed to be addressed for the article to be suitable for publication.

Thank you for the comment that the revision has clarified key technical aspects of the manuscript.

My conclusion is that this is solid work. However, the authors cannot claim they have done a quantum simulation. There is definitely evidence that tunnel coupling impacts the Coulomb blockade transport in this system. If the authors modify the title, abstract, and claims of the paper to avoid over-claiming 2D quantum simulation, and focus on claims that can be justified in Coulomb blockade transport, then the article would most likely be suitable for publication.

We have modified the title, abstract, and claims of the paper to avoid over-claiming 2D quantum simulations. We have removed all explicit references to demonstration of an analog quantum simulator. In light of the above comments regarding tuning of parameters in the Hamiltonian, we have made the following changes:

We have modified the title to "Experimental Realization of an Extended Fermi-Hubbard Model Using a 2D Lattice of Dopant-based Quantum Dots" and removed the words "Quantum Simulation" from the title.

We have removed "Here, we overcome these challenges by integrating the latest developments in atomic fabrication and demonstrate the analog quantum simulation of a 2D extended Fermi-Hubbard Hamiltonian using STM-fabricated 3x3 arrays of single/few-dopant quantum dots" from the abstract.

In the introduction the wording now reads "To augment the interpretation of our simulated extended Hubbard Hamiltonian, we numerically solve the Hubbard Hamiltonian..." which replaced the wording "To augment the interpretation of the analog quantum simulations, we numerically solve the Hubbard Hamiltonian".

P. 17 now reads "In this first experimental realization of an extended Fermi Hubbard Hamiltonian using atomically patterned 2D arrays of dopants" instead of "In this first demonstration of an analog quantum simulation of an extended Fermi Hubbard Hamiltonian using atomically patterned 2D arrays of dopants"

P 18 in the Discussion Section we have removed "We anticipate that dopant-based analog quantum simulation experiments using larger arrays should soon be able to provide answers not available to numerical calculations."

There are a few additional examples in the text where we made similar changes to replace the analog quantum simulation wording. These are identified in the version showing track changes.

Reviewer #3 (Remarks to the Author):

The authors have successfully changed the manuscript such that my comments and concerns have been addressed. I am happy for publication to proceed.

We thank the reviewer for your time and helpful comments.